# TMEM16F and dynamins control expansive plasma membrane reservoirs

Christine Deisl [1,4], Donald W. Hilgemann [1,4✉], Ruhma Syeda[2] & Michael Fine [1,3✉]

Cells can expand their plasma membrane laterally by unfolding membrane undulations and by exocytosis. Here, we describe a third mechanism involving invaginations held shut by the membrane adapter, dynamin. Compartments open when Ca activates the lipid scramblase, TMEM16F, anionic phospholipids escape from the cytoplasmic monolayer in exchange for neutral lipids, and dynamins relax. Deletion of TMEM16F or dynamins blocks expansion, with loss of dynamin expression generating a maximally expanded basal plasma membrane state. Re-expression of dynamin2 or its GTPase-inactivated mutant, but not a lipid binding mutant, regenerates reserve compartments and rescues expansion. Dynamin2-GFP fusion proteins form punctae that rapidly dissipate from these compartments during TMEM16F activation. Newly exposed compartments extend deeply into the cytoplasm, lack numerous organellar markers, and remain closure-competent for many seconds. Without Ca, compartments open slowly when dynamins are sequestered by cytoplasmic dynamin antibodies or when scrambling is mimicked by neutralizing anionic phospholipids and supplementing neutral lipids. Activation of Ca-permeable mechanosensitive channels via cell swelling or channel agonists opens the compartments in parallel with phospholipid scrambling. Thus, dynamins and TMEM16F control large plasma membrane reserves that open in response to lateral membrane stress and Ca influx.

[1] University of Texas Southwestern Medical Center, Department of Physiology, Dallas, TX, USA. [2] University of Texas Southwestern Medical Center, Department of Neuroscience, Dallas, TX, USA. [3] University of Texas Southwestern Medical Center, Department of Molecular Genetics, Dallas, TX, USA. [4] These authors contributed equally: Christine Deisl, Donald W. Hilgemann. ✉email: Donald.hilgemann@utsouthwestern.edu; Michael.Fine@utsouthwestern.edu

Most cells expand their plasma membrane (PM) laterally to prevent tears during mechanical perturbation, cell swelling and spreading, and macropinocytosis and phagocytosis[1–4]. Since phospholipid (PL) bilayers can be stretched by only 4% without microlysis[5,6], expansion is assumed to rely on unfolding of PM undulations (i.e., ruffles and micro-folds) and exocytosis of new membrane. PM unfolding typically occurs first[4,7], while exocytic events occur as PM tension increases and cytoplasmic Ca rises[4,6–8]. Here, we describe that, in addition, large PM reservoirs are electrically and functionally isolated from bulk PM via the adapter protein, dynamin[9], which oligomerizes around PM invaginations[10] as it binds anionic PLs[11]. During lateral PM stress, Ca influx by mechanosensitive channels activates the PL scramblase, TMEM16F[12], resulting in the exchange of cytoplasmic anionic PLs for neutral PLs, followed by dynamin relaxation and opening of the reserve PM compartments. The highly distinct feature of this mechanism is that large PM reserves are reversibly isolated without being excised from the PM.

PM expansion can be monitored via fluorescent fusion proteins, reversible PM labeling with dyes[7,13], and electrical recordings of PM capacitance ($C_m$)[14,15]. Regardless of the method, PM expansion can be 10-fold greater than available secretory vesicle pools[16], and individual $C_m$ steps can correspond to many square microns of membrane[15]. This suggests involvement of membrane networks, and not surprisingly, expansion is insensitive to tetanus toxins that block exocytosis in neurons[17]. Previous suggestions are that PM expansion during cell spreading involves membrane derived from Golgi[7], and that PM expansion during cell wounding reflects lysosome exocytosis[18].

Recently, we described that Ca-dependent PM expansion is coupled to the activity of the ion channel and PL scramblase[17], TMEM16F, also known as ANO6. TMEM16F mutations decrease platelet pro-coagulant activity and modify T-cell and glia function[19]. TMEM16F is widely expressed, exceptions being cardiac myocytes and most neurons[20]. It mediates PL scrambling during platelet activation but not during apoptosis[21]. Ion selectivity of TMEM16F varies with the degree of Ca-activation[22], and we previously reported that blockade by cytoplasmic polyamines inhibits both phosphatidylserine (PS) scrambling and PM expansion[17]. Mutations that alter channel function but preserve modest PL scrambling rescue expansion[17]. Thus, scrambling of PLs is critical for PM expansion.

In this work, we elucidate the basic mechanisms by which PL scrambling leads to PM expansion by accessing a membrane reserve and reveal one role for this reserve in mechanically induced membrane stress. To address how scrambling activates expansion, we pursued a lead that many membrane adapters bind anionic PLs[23]. We describe now that the endocytic adapter, dynamin2[24], isolates PM reserves and prevents their opening until a loss of cytoplasmic phosphatidate (PA), PS[11,25], and phosphoinositides (PIP$_2$)[11] allows them to relax. Further, we show that cells activate this process during both cell swelling and activation of Ca-permeable mechanosensitive channels[26], revealing a previously undescribed cellular mechanism in the regulation and relief of lateral PM stress.

## Results

### TMEM16F and dynamins control an extensive PM reserve.
Figure 1 documents the dual dependence of Ca-activated PM expansion on TMEM16F and dynamins. Figure 1A illustrates the parallel recording of PM area as $C_m$[15] and PS exposure as binding of rhodamine-heptalysine (K7r)[16,27] in HEK cells, as in many other cell types[17] (Supplementary Figs. 1 and 2). Brief Ca influx by a Ca-ionophore, ionomycin, or by reverse Na/Ca exchange[16] (BHK-NCX cells) increases cytoplasmic Ca, activating TMEM16F

and promoting PM expansion by >50% in 5–30 s (Fig. 1A, Supplementary Figs. 1 and 2B). Concurrent imaging reveals threefold increases in extracellular K7r labeling as PLs scramble. As described[17], extended ionophore exposure can cause extensive PM shedding after PM expansion (Supplementary Vid. 1, Supplementary Fig. 2C). Verifying that TMEM16F activity is coupled to PM expansion[16,17], TMEM16F ablation by CRISPR (TMEM16F-null)[17], blockade by cytoplasmic spermine, and natural lack of endogenous expression (e.g., Neuro2A[28] and murine cardiomyocytes[20,30]) all abrogate K7r binding and PM expansion (Fig. 1A, Supplementary Fig. 1). Ca influx then leads to rapid PM internalization by massive endocytosis (MEND)[17].

Figure 1B–F delineate dynamin2 function in TMEM16F-dependent PM expansion. Dynamins are cytosolic GTPase adapters that bind PM-specific PLs and promote endocytosis[24]. Their affinity for anionic PLs and the ability to constrict membranes suggested a potential role in PM expansion. Often, dynamins appear as discrete PM-associated punctae[29]. In TMEM16F-null HEK cells and cells rescued with mTMEM16F, WT-Dnm2-GFP expression initially shows similarly extensive PM punctae (Fig. 1B, left). During ionomycin exposure, however, cells rescued with TMEM16F (Fig. 1B, top) dramatically lose Dnm2 punctae while cells lacking TMEM16F (Fig. 1B, bottom) robustly retain them. Impressively, WT-Dnm2 and the GTPase-dead dominant-negative K44A Dnm2 dissociate similarly. However, TMEM16F activation fails to dissociate the lipid-binding K562E Dnm2 mutant[30] (Fig. 1B).

Three dynamins, Dnm1, 2, and 3, regulate adapter-dependent endocytosis[24]. To determine their roles in PM expansion we employed MEF cells with inducible knockout of all three dynamins[31] via Cre expression or tamoxifen exposure, as well as non-small cell lung adenocarcinoma cells (H1299) lacking Dnm1[32] and allowing further deletion of Dnm2[33]. To trigger PM expansion, we employed pipette (cytoplasmic) solutions with free Ca heavily buffered with EGTA to 7 μM[17] with comparable results observed for ionomycin (Supplementary Fig. 2B). As illustrated in the upper left panel of Fig. 1C, uninduced Dnm TKO cells expanded by >50%, similar to BHK, HEK, and Jurkat cells (Supplementary Fig. 1), as well as H1299 cells lacking Dnm1[32] (Fig. 1D). Conversely, when dynamins are ablated in tamoxifen-treated cells, PM fails to expand at all. Expansion fails similarly with Cre-induction or using H1299 cells with additional Dnm2 deletion[33] (Fig. 1D). In induced Dnm TKO cells, rescue with either WT-Dnm2 or GTPase-inactive K44A Dnm2[34] recovers PM expansion, while the lipid-binding mutant K562E does not. In summary, Dnm2 specifically rescues TMEM16F-dependent PM expansion. In contrast to dynamin function during endocytosis[25], Dnm2 GTPase activity is not required for compartment formation or PM expansion. However, strong dynamin interactions with anionic lipids, lost in the K562E mutant[30], are essential.

Since cells employed here are spheroidal during patch clamp, we could calculate the percent excess PM area (PM$_{excess}$) from the ratio of spheroidal area ($A_{smooth}$), determined from micrographs, to real PM area measured as $C_m$ ($A_{Cm}$ with 1 pF≡110 μm$^2$)[35] (Fig. 1C, bottom left; PM$_{excess}$ = 100*($A_{Cm}/A_{smooth}$-1)). At 37 °C, perfusion of 7 μM free Ca induced expansion and caused excess PM area to increase from ~170 to >400% in uninduced Dnm TKO cells. Impressively, the initial excess PM area amounted to >500% in induced Dnm TKO cells and did not increase further with Ca. Although an increase in PM area after dynamin deletion is consistent with loss of endocytosis, subsequent experiments suggest it reflects the constitutive opening of Dnm-constricted PM domains.

The right panels of Fig. 1C describe how Dnm2 mutants modify resting PM. Initial resting excess PM area (Excess Basal

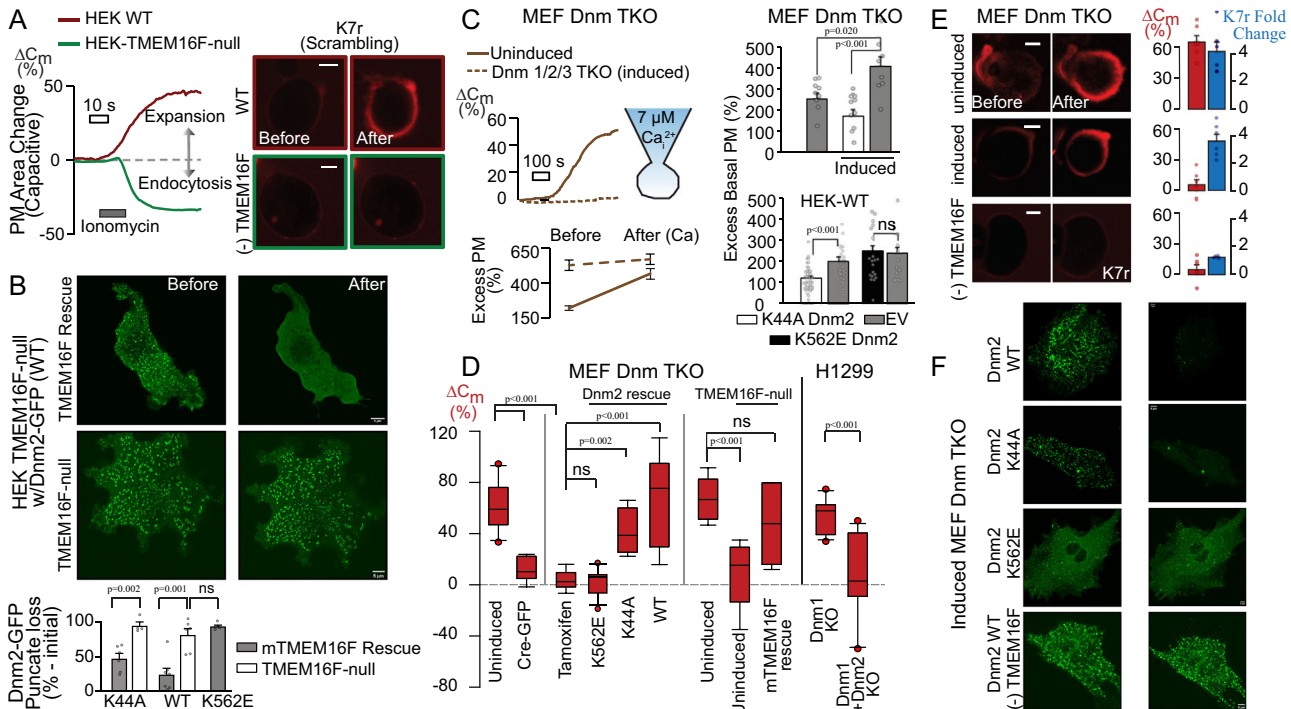

**Fig. 1 Dynamin plays key roles in TMEM16F-dependent PM expansion. A** Representative recordings of PM area changes (%$\Delta C_m$) and PS scrambling in HEK293 cells with or without TMEM16F. All recordings at 37 °C, unless stated otherwise. Confocal images show extracellular binding of the fluorescent cationic peptide, K7r, used to track PS exposure by TMEM16F, $n = 10$. **B** Dynamin PM localization is disrupted by TMEM16F activation. Representative live-cell super-resolution airyscan confocal images of Dnm2-GFP in HEK-TMEM16F-null cells (bottom) and rescued with mTMEM16F (top) before and after Ca$^{2+}$/ionomycin treatment. Images created from 1 µm z-stack near the adhesion surface. Composite results below show Dnm2 punctae loss for Dnm2 WT and K44A mutant but not K562E or when TMEM16F is absent, ionomycin: 5 µM; $n = 6, 4, 7, 6, 4$, resp. **C** Left, Ca$^{2+}$ dialysis increases PM area (%$\Delta C_m$) and excess PM (1 pF = 110 µm$^2$*$C_m$)/($\pi$*d$^2$) in uninduced MEF Dnm TKO cells but is blocked when expression of all three Dnm isoforms is ablated (tamoxifen induction) as excess PM remains stable and elevated, $n = 6$. Right. Dnm2 expression can control basal excess PM. The resting excess basal PM area was determined without elevated Ca and at RT in Dnm TKO and HEK cells. Cells were either transfected with mCherry (EV) or co-transfected with mCherry and K44A or K562E Dnm2. Induced cells lacking Dnm had increased basal excess PM while cells overexpressing K44A but not K562E reduced excess membrane, $n = 9, 8,11,42,33,20,11$. **D** Composite results for PM expansion in MEF Dnm TKO and H1299 cells (7 µM free Ca), $n = 10, 6, 6, 13, 4, 9, 9, 9, 5,18,12$. **E** Micrographs and composite results for K7r binding (scrambling) in uninduced (top), induced (mid), and uninduced TMEM16F-null cells (bottom), $n = 6, 6, 6, 6, 6, 4$. **F** As in (**B**), representative images of punctae loss in Dnm2-rescue (WT, K44A, K562E, and WT/TMEM16F-null) in induced MEF Dnm TKO cells, $n = 4$ independent experiments. All data analyzed from the total number of independent cells ($n$) from a minimum of three experiments and expressed as mean ± s.e.m. Unpaired Student's $t$-test used for comparing two groups. All Scale bars: 5 µm. Box plot displays median value and upper and lower quartiles with whiskers representing 10th/90th percentile.

PM) was calculated for cells at 24 °C without Ca. As expected, cells lacking dynamins had twice as much initial excess PM as did uninduced cells or induced cells rescued with K44A Dnm2. Similarly, excess basal PM in HEK cells co-expressing the K44A Dnm2 and fluorescent mCherry was reduced >40% compared to cells expressing mCherry alone. In contrast, the excess area did not decrease in cells expressing K562E Dnm2. Thus, dominant-negative K44A Dnm2 and dynamin deletions generate very different phenotypes with respect to PM expansion. K44A Dnm2 generates reserve compartments and supports PM expansion, while dynamin deletions constitutively expand the PM. This is very different for endocytosis phenotypes in which K44A Dnm2 and dynamin deletions have similar blocking effects. Importantly, the generation of tubular membrane structures was noted previously for K44A Dnm2[34].

We next illustrate TMEM16F function in dynamin-dependent PM expansion. PM expansion and PL scrambling, detected with K7r, occur in parallel in uninduced Dnm TKO cells. Induced cells lacking dynamins fail to expand but maintain PL scrambling, while cells lacking TMEM16F fail to both expand *or* scramble (Fig. 1E). Similar to HEK cells, both induced and uninduced Dnm

TKO cells lacking TMEM16F undergo MEND in response to Ca[36]. As described for HEK cells, Dnm TKO cells rescued with Dnm2-GFP and related mutants also display discrete dynamin punctae (Fig. 1F and Supplementary Fig. 3). During Ca-ionomycin treatment, WT and K44A Dnm2 punctae disperse almost completely, while K562E Dnm2 punctae remain. Dnm TKO cells lacking TMEM16F also fail to disperse dynamin puncate. This is further illustrated in super-resolution live-cell imaging that reveals Dnm2 punctae as submicron rings that dissipate during PL scrambling (Supplementary Fig. 3).

A clear limitation of knockout experiments is their long-term nature. As an acute approach, we demonstrate in Supplementary Fig. 2D that high-affinity dynamin-binding SH3 domains from amphyphysin2 (Amph-SH3)[37,38] block PM expansion during Ca elevation in WT Jurkat cells and BHK cells. This is consistent with Amph-SH3 domains stabilizing critical oligomeric dynamin structures in vivo[39], presumably by binding to multiple dynamin sites that engage partners[40], whereas these domains can dissipate dynamin rings in vitro[38]. In summary, the PM expands constitutively to a maximal extent after knock-out of all three dynamins. This phenotype is rescued by either WT Dnm2 or

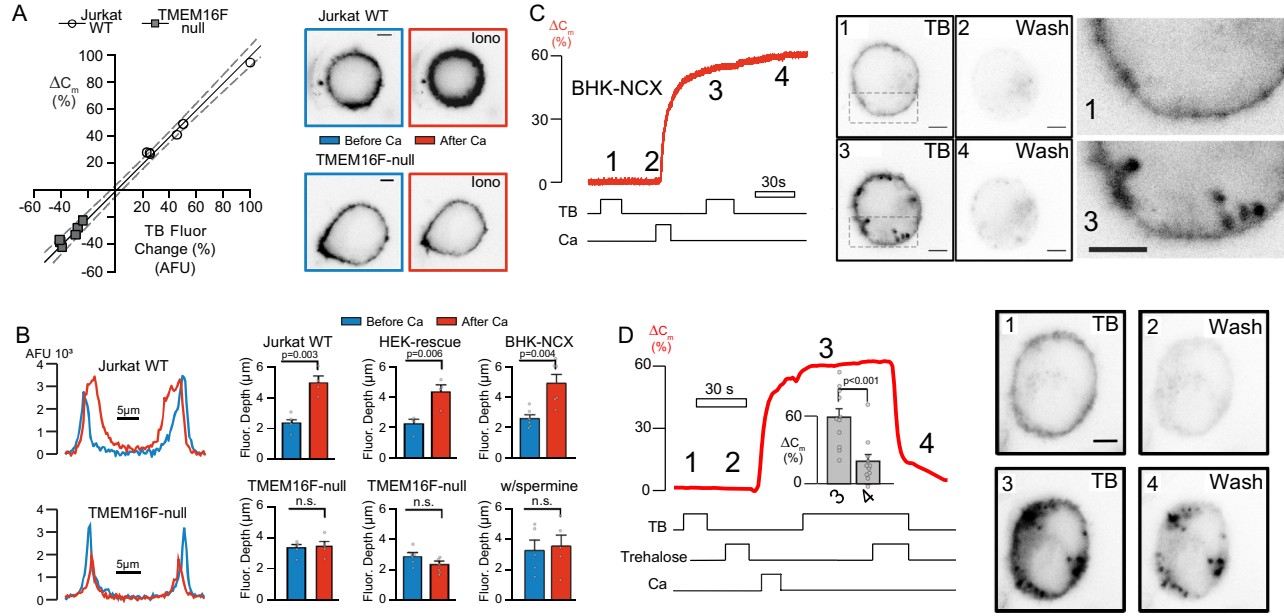

**Fig. 2 Ca elevation opens compartments that extend deeply into the cytoplasm. A** Reversible fluorescent PM staining by TB before and after Ca elevation in WT and TMEM16F-null Jurkat cells reveal TB fluorescence as a relatively uniform indicator for PM changes. Changes of TB fluorescence (%) are plotted against changes of $C_m$ (%) in response to Ca elevation. Linear regression of results with 95% C.I. **B** Representative fluorescence line scans during TB application in Jurkat WT and TMEM16F-null cells before (blue) and after (red) Ca elevations. Quantification of average PM fluorescence depth in WT ($n = 4$) and TMEM16F-null Jurkat cells (5), TMEM16F-null HEK cells with (4) and without (6) rescued TMEM16F expression and BHK cells with (5) and without spermine (6). Confocal imaging of a BHK cell during TB application and after washout, before and after Ca elevation via NCX1. Extracellular continuity of labeled compartments (3) is indicated by washout (4) **D** As in (**C**) Trehalose exposure at low ionic strength reverses PM expansion (4), $n = 11$. All data were analyzed from the total number of independent cells ($n$) from a minimum of two experimental repeats and expressed as mean ± s.e.m. Paired Student's t-test was used for comparing two groups for (**B**) and (**D**). All Scale bars: 5 μm.

GTPase-inactive K44A Dnm2 but not the K562E Dnm2 lipid binding mutant. Triple dynamin loss, as well as loss of Dnm1 and 2, but not Dnm1 alone, blocks TMEM16F-mediated PM expansion. Punctae formed by WT and K44A Dnm2 dissipate during PL scrambling, while punctae formed by K562E Dnm2 do not. Dynamin-binding Amph-SH3 domains acutely block PM expansion. In summary, dynamin restrictions generate labile PM reservoirs that, in response to PL scrambling, open and dramatically expand the PM.

**Identification of the PM compartment opened by TMEM16F and dynamins**. Next, we investigated sources of membrane expanding the PM. PS labeling reveals that newly exposed PM can extend deeply into the cytoplasm (Supplementary Figs. 5, 6 and Supplementary Vid. 2). For quantitative analysis, we exploited Trypan Blue (TB) fluorescence that occurs upon its uniform binding to membrane sugars and proteins[41,42]. In contrast to K7r and FM dyes[17], it does not intercalate into bilayers and is unaffected by anionic PLs. Its red fluorescence at the PM[41] remains proportional to $C_m$ during PM expansion as well as MEND (Fig. 2A; Pearson correlation coefficient, 0.9; slope, 0.96 ± 0.03). Thus, TB reports PM area more accurately than other dyes. As shown in Fig. 2B, cells were patched and labeled with TB, followed by washout. Intracellular Ca was elevated by Ca influx via ionomycin or NCX1, and after PM expansion cells were relabeled with TB. Ca influx clearly promotes PM labeling extending several microns into cells. Line scans reveal that average staining depths, initially 2–3 μm (blue in Fig. 2A/B), increase by several microns after Ca influx (red in Fig. 2A/B, top). As expected, TMEM16F-null cells and BHK-NCX cells with spermine show no change (Fig. 2A/B).

Compared to Jurkat and HEK cells, BHK cells have larger cytoplasmic spaces and smaller nuclei, thereby allowing better

visualization of PM invaginations. As shown in Fig. 2C and Supplementary Vid. 3, TB fluorescence reveals extensive reversible labeling of BHK PM invaginations with depths >1 μm. Rapid wash-off of TB from new compartments is demonstrated in Fig. 2C. Image 4 documents that newly opened compartments remain open and separate from the cytoplasm. Importantly, invaginations appear constricted via neck-like structures at the outer cell surface, as expected for dynamin binding[43]. To improve membrane resolution, control and ionomycin-treated cells were rapidly fixed, labeled with TB, and imaged at super-resolution, producing three-dimensional Z-projections across the cell. Supplementary Vid. 4 and 5 detail both compartment depth and PM restriction points in control and ionomycin-treated cells, respectively. High-resolution live-cell analysis of BHK cells also reveals that initial Dnm2-GFP punctae occurring proximal to the adhesion surface align with newly exposed TB after compartment opening (Supplementary Fig. 4). Again, compartment dimensions are consistent with large $C_m$ steps during PM expansion[15] and inconsistent with classical exocytosis.

Given the stability of compartment morphology in the absence of PM tension, we tested whether invaginations could reclose by promoting membrane-membrane interactions. Disaccharides such as trehalose decrease the mobility and hydration of PLs[44]. As shown in Fig. 2D, PM expansion indeed reverses up to 30 s after opening when 90% of extracellular ions are replaced iso-osmotically by trehalose (Fig. 2D). TB staining then does not wash off, and a normal ionic solution does not reopen compartments (Fig. 2D, Image 4 vs. Fig. 2C, Image 4), indicating that newly opened compartments resealed and trapped TB. These patterns are again inconsistent with classical exocytosis in which fusion pores are reversible only briefly[45]. Furthermore, the patterns are inconsistent with $C_m$ changes arising from scrambling-induced changes of the PM dielectric. First, the

trehalose effects are too fast, and second, PL scrambling occurs without $C_m$ changes in induced Dnm TKO cells (Fig. 1E).

Next, we address whether PM expansion involves organelle fusion with the PM[46,47]. In brief, labeling with lysosome-specific probes and expression of fluorescent protein markers for ER, lysosomes, and trans-Golgi provided no evidence for significant organelle involvement (Supplementary Fig. 7A–E, Supplementary Vid. 6). Neither colocalization nor fluorescent quenching strategies supported a presence of organellar probes in newly exposed PM. While we cannot discount small organellar contributions, the magnitude of PM expansion clearly requires the involvement of tubules and/or membrane networks. We tested whether recycling endosomes contribute to PM expansion using well-established FM dye protocols[13]. In attached WT Jurkat cells, only a minor loss of FM-labeled endosomal particles occurred (Supplementary Fig. 7F–G). The numbers of recycling endosome fusion events, however, fell far short of accounting for PM expansion. Other groups have described the existence of a large organelle, dubbed enlargeosome[48], opening in the PM during Ca elevations. Enlargeosome opening was described to expose the bulk endocytic marker, VAMP4. We confirm that heterologously expressed VAMP4, but not VAMP2, becomes exposed during PM expansion (Supplementary Fig. 8, Supplementary Vid. 7). In addition, super-resolution imaging of live cells expressing WT-Dnm2-mRuby3 co-expressed with either VAMP2 or VAMP4 reveals that VAMP4, but not VAMP2, correlates with Dnm2 punctae (Supplementary Fig. 9). These results suggest that enlargeosomes open by the mechanisms delineated here.

**Ca-independent PM compartment opening and closing.** TMEM16F-dependent PM expansion occurs slowly with free Ca in a normal signaling range of 0.3 to 1 micromolar (Supplementary Fig. 2A). This is expected because TMEM16F-dependent expansion has only shallow dependence on cytoplasmic free Ca[16], in contrast to synaptotagmin-mediated exocytosis[49]. At the next level, we asked whether PM reservoirs might open independent of Ca or TMEM16F via interventions that mimic PM changes during PL scrambling and/or modify dynamin-membrane interactions.

As shown in Fig. 3A, uninduced Dnm TKO cells exhibit substantial PM expansion during dialysis at 24 °C with reagents that chelate anionic PLs, including a cationic RAF-1 peptide that binds PA selectively (PA peptide)[50] (Fig. 1A). Similar results were observed in BHK and Jurkat cells for PA peptide, protamine and K7r (Supplementary Fig. 10A/B/C). Pipette solutions (cytoplasmic) were ATP-free and contained 10 mM EGTA or EDTA to aggressively chelate Ca and, in the case of EDTA, also Mg. Under these conditions, sequestration of anionic PLs causes moderate PM expansion of 10–20% of PM area, albeit at a slower rate than with Ca. These modest responses are advantageous to visualize $C_m$ noise and step-wise behaviors (Supplementary Fig. 10A), indicating expansion via stochastic events rather than PM dielectric changes. Results suggest further that multiple anionic lipids support compartment closure. EDTA promoted a small expansion in comparison to EGTA (Fig. 3A and Supplementary Fig. 10C), suggesting that a Mg-dependent process may be important, inhibition of Mg-dependent lipid kinases being one possibility. Mg-independent phosphatases, which are known to be highly active in the PM[51], would promote dephosphorylation of PA and PIP2 after Mg chelation. These phosphatases are inhibited by orthovanadate ($VO_4$)[51], and as expected, the inclusion of $VO_4$ (1 mM) in the pipette solution blocked (Fig. 3A) the small EDTA-induced expansions. Further supporting a role for PA, in the presence of $VO_4$ both 1 and 10 μM of the PA-binding peptide

rescued this block, and PM expansion increased further with $VO_4$-free dialysis solutions. Related to this result, cleavage of PIP2 during activation of hM1 muscarinic receptors causes small but significant PM expansions in BHK cells[16]. Importantly, PS, PA, and PIP2 are all bound by dynamins[11,25] and PA supports dynamin penetration of bilayers just as strongly as PIP2[11]. As expected, PM changes remained <5% during dialysis of 10 μM PA-binding peptide in induced Dnm TKO cells (Fig. 3A), even in $VO_4$ free conditions. Thus, PM expansion during sequestration of anionic PLs, while moderate in magnitude, is clearly Dnm-dependent.

Since PM expansion by cationic peptides is relatively modest, we tested whether PM changes besides loss of anionic PLs might support PM expansion. PL scrambling is non-selective and bidirectional[52], causing both a loss of cytoplasmic anionic PLs and a gain of neutral PLs, such as phosphatidylcholine (PC)[52]. To examine whether PC accumulation might be important, we included labile PC liposomes in pipette solutions, together with 0.3 mM albumin to promote PL exchange[53]. As shown for uninduced Dnm TKO cells (Fig. 3A, right), albumin alone was without effect, as was PS supplementation. PC supplementation, however, resulted in 23% PM expansion, and this expansion was significantly reduced in cells lacking dynamin. Thus, dynamins are required for PC-mediated expansion. Both Jurkat and BHK cells behave similarly to MEF cells with interventions mimicking PL scrambling (Supplementary Fig. 11). In Jurkat cells, however, PM expansion with protamine alone and neutral lipid supplementation alone was not significant. However, when both protamine and albumin/PC were dialyzed to simulate scrambling, expansion was 28%. Evidently, PM biophysical changes besides a loss of anionic PLs are important in the final triggering of dynamin-dependent PM expansion.

Since PL manipulations can induce PM expansion without Ca, we also tested whether dynamin manipulations might induce PM expansion without Ca. As shown in Fig. 3B, pipette solutions containing Dnm2 antibodies, dynamin inhibitory peptide (DIP), or Dynab, a functional dynamin1/2 nanobody[54], each generate robust PM expansion over 10 min without Ca. Buffer controls for antibodies and peptide solutions, dialysis of Dnm1, an unrelated Brain-Natriuretic peptide (BNP), and FITC-labeled secondary antibodies are all ineffective (Fig. 3B). Fluorescent secondary antibodies clearly enter cells over 3 min with no effect on PM area (Supplementary Fig. 11D). Accordingly, disruption of dynamin-PL interactions can clearly promote compartment opening. Together, these results solidify the central role of dynamin2 in PM expansion. We now connect this mechanism to physiologically relevant lateral membrane stress that occurs in cell swelling and mechanosensation, as sketched in cartoon form in Fig. 3C.

**Mechanosensitive cation channels trigger physiological PM expansion.** As noted in the Introduction, cell swelling and spreading require that the PM expands extensively, and cell swelling triggers PM expansion via Ca-influx through mechanosensitive cation channels[55–57]. As shown in the left panel of Fig. 4A, Jurkat WT cells expand by 20% during application of 50% hypo-osmotic solution when extracellular Ca is present, while Jurkat TMEM16F-null cells undergo endocytic responses. Using BHK cells (right bar graphs in Fig. 4A), PM expansion in hypotonic solution amounts to ~40%. These responses are blocked by Ca chelation with 5 mM EGTA, and MEND occurs when TMEM16F is blocked by 1 mM cytoplasmic spermine, similar to responses described in Supplementary Figs. 1 and 2.

To address the physiological nature of these responses, we analyzed cytoplasmic Ca and PS translocation in un-patched WT and TMEM16F-null Jurkat cells using the red Ca indicator, X-

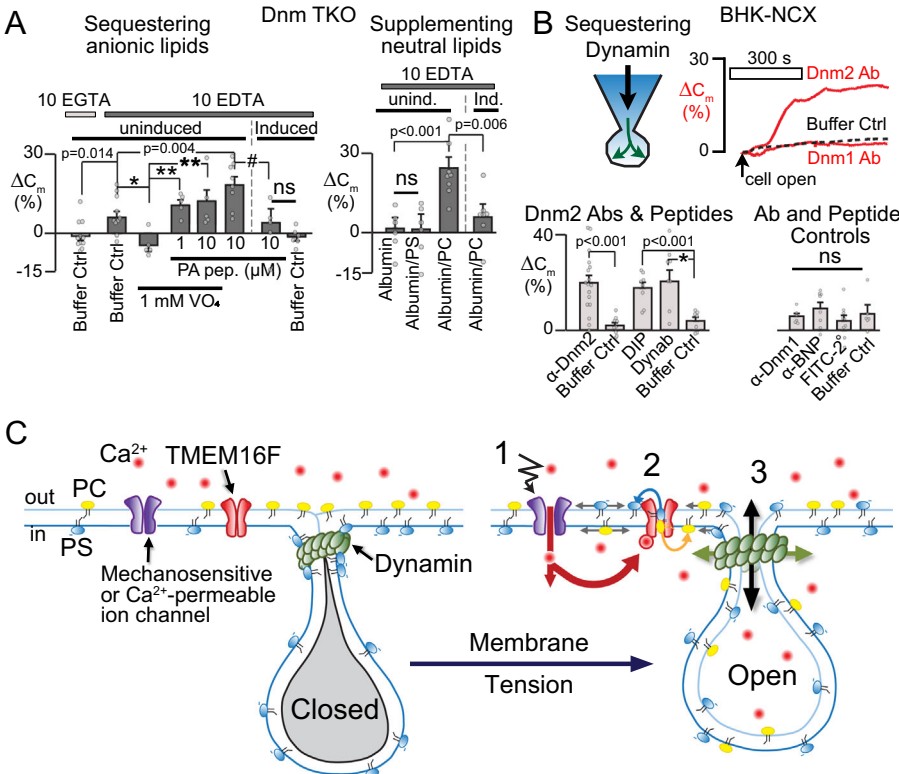

**Fig. 3 Manipulations of PLs that mimic scrambling can trigger PM expansion without Ca or TMEM16F activation. A** Sequestration of inner leaflet anionic PLs induces modest PM expansion. Dnm TKO cells with cytoplasmic buffer supplemented as follows: 0 ATP, 0 Ca, and 10 mM EGTA or EDTA, as indicated; PA Peptide 10 or 1 µM, and 1 mM orthovanadate ($VO_4$). PM expansion induced by PA-binding peptide is significantly reduced in induced Dnm TKO cells lacking dynamins (*$p = 0.008$; **$p < 0.001$; #$p = 0.020$). Supplementation of cytoplasm with albumin or albumin/PL (0.2/0.2 mM) mix in Dnm TKO cells (right). The anionic PL, PS, and albumin alone do not induce PM expansion; however, supplementation with neutral PC induced PM expansion in the absence of Ca and TMEM16F activation. Expansion is blocked in induced Dnm TKO cells lacking dynamins, $n = 11,11,5,5,5,8,7,6,6,8,7$. **B** Composite results of BHK cells in Ca-free (10 mM EGTA) conditions over 10 min during cytoplasmic dialysis with Dnm1 and Dnm2 antibodies, dynamin inhibitory peptide (DIP) 100 µM, Dynab nanobody, and relevant controls (brain-natriuretic peptide and Dnm 1 antibody, FITC-labeled goat secondary antibody (*$p = 0.003$), $n = 17,9,8,8,7,6,8,9$. **C** Working model for the physiological function of dynamin-constricted PM compartments. (1) Membrane tension activates Ca-influx via mechanosensitive cation channels of multiple types. (2) A local elevation of cytoplasmic Ca activates TMEM16F and PL scrambling with loss of anionic PLs (yellow) and gain of neutral PLs (blue) in the inner monolayer. (3) In response to PL changes, dynamin constrictions (GREEN) relax and compartments open, allowing the sequestered membrane to alleviate lateral PM stress. All data were analyzed from the total number of independent cells ($n$) from a minimum of two experiments and expressed as mean ± s.e.m. Unpaired Student's $t$-test was used for comparing two groups.

Rhod-1, and the PS-specific stain, Annexin V-FITC. Brief incubation times of 5 min and relatively gentle 50% hypotonic conditions were used to avoid lytic responses that might generate PL scrambling via PM disruptions, rather than scramblase activity. WT and TMEM16F-null Jurkat cells exhibited similar transient Ca elevations in similar fractions of all cells (Fig. 4B, bottom right), indicating that TMEM16F is not the primary osmosensor and source of Ca influx during cell swelling[55]. However, WT cells exhibited a nearly threefold increase in Annexin-V labeling, while TMEM16F-null cells showed no response (Fig. 4B, bottom left), demonstrating that PL scrambling during swelling is triggered by TMEM16F and likely initiates PM expansion responses. Thus, Ca influx during lateral membrane tension can promote compensatory TMEM16F-dependent PM expansion. Although the specific ion channels that mediate Ca influx during cell swelling remain to be identified, mechanosensitive channels, such as PIEZO1[56], have been shown to activate Ca influx in response to lateral PM stress in many cell types[56,58]. Therefore, we exploited PIEZO1 to test whether a defined mechanosensitive channel can initiate PM expansion. PIEZO1 channels, expressed in HEK TMEM16F-null cells[17] and mTMEM16F rescued cells, were stimulated with the PIEZO1-specific agonist Yoda1[59]. In the presence of extracellular Ca, large

PM expansions occurred in cells expressing both TMEM16F and PIEZO1 (left bar chart in Fig. 4C). WT HEK293 cells lack sufficient PIEZO1 to respond (center bar chart in Fig. 4C), and TMEM16F-null cells with PIEZO1 undergo MEND (right bar chart in Fig. 4C). Complete responses during Yoda1 are shown in Fig. 4D. Together, the results demonstrate that PIEZO1 can initiate local Ca transients and PL scrambling that would physiologically activate PM expansion during mechano-perturbation.

## Discussion

This study delineates a molecular pathway in which PL scrambling by TMEM16F initiates large-scale PM expansion in diverse cell types. A key question is whether PM expansion involves SNARE proteins and bona fide membrane fusion events[60]. While it is suggested that dynamins can act as a filter for SNARE function[61], our results would require dynamins to directly inhibit fusion events until PLs scramble. Furthermore, the SNAREs would have to be tetanus toxin-insensitive[15–17], and any fusion events would have to remain reversible for an extended period to explain closure with trehalose (Fig. 2D), a pattern not reported for SNARE-dependent fusion events. As outlined in Fig. 3C, the alternative is

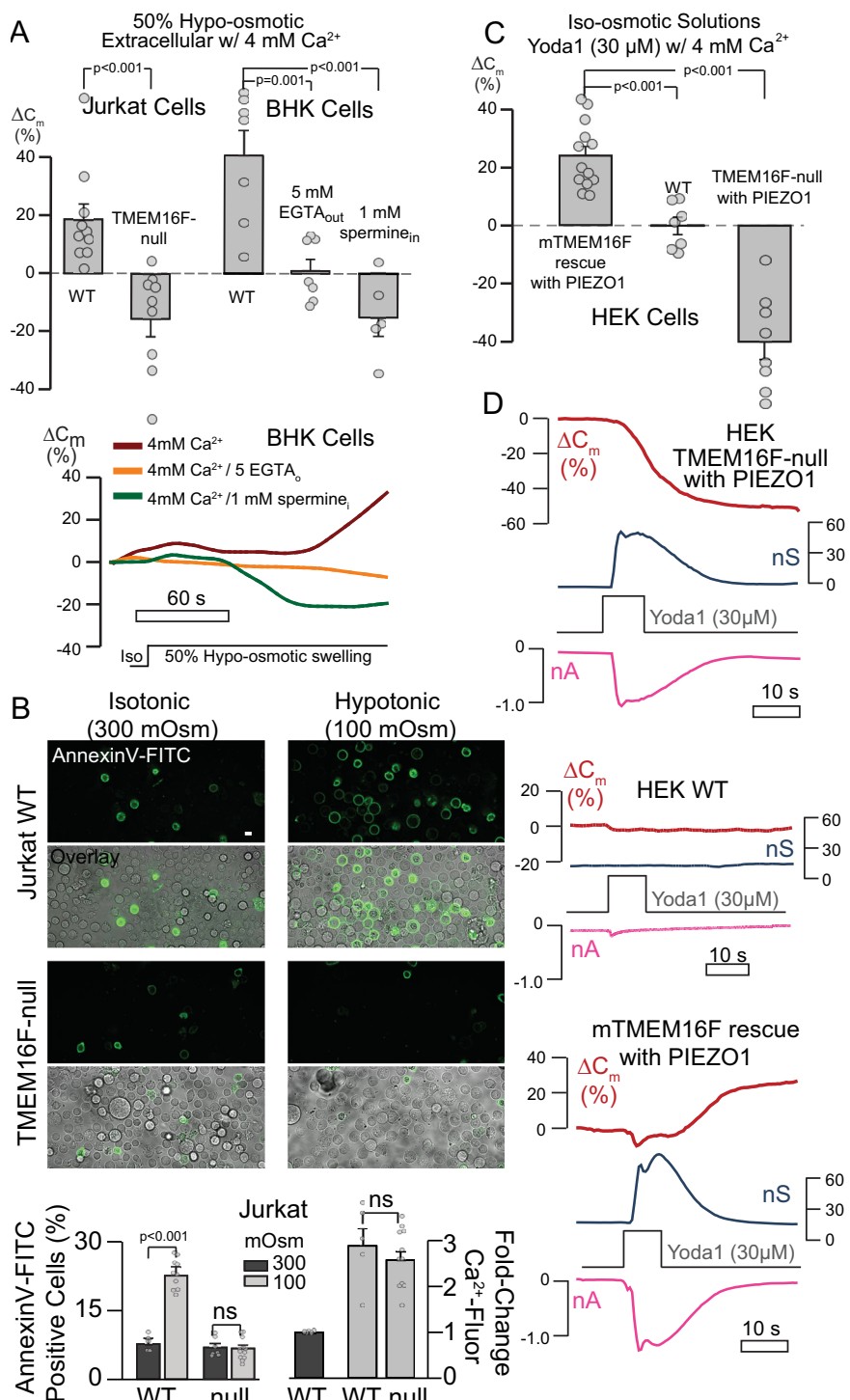

**Fig. 4 Physiological proof of principle: mechanosensitive opening of TMEM16F-dependent PM reservoirs. A** PM area changes in whole-cell recordings of Jurkat WT, TMEM16F-null, and BHK WT cells subjected to hypo-osmotic stimulation with 4 mM extracellular Ca. Below, representative traces for BHK cells, $n = 10,9,7,7,5$ independent cells from three experiments. **B** Annexin V-FITC labeling (top) to label surface PS exposure and bright-field overlay (bottom) of Jurkat WT and TMEM16F-null cells in isotonic and hypotonic solutions with 4 mM Ca. Sucrose was used to adjust osmolarity. Quantification below with percentage of AnnexinV positive cells (left) and fold increase in Ca-fluorescence (right) in cells loaded with X-Rhod1 (3 μM) Ca-indicator normalized to initial Ca values indicating that hypotonic-induced Ca-entry is independent of TMEM16F expression, $n = 5,10,7,12,5,5,11$ biologically independent samples. **C** Quantification of $C_m$ changes during application of 4 mM Ca and PIEZO1 agonist Yoda1 in HEK-TMEM16F-null cells overexpressing PIEZO1 (right) and rescued with mTMEM16F (left) as well as WT HEK293 cells (middle), $n = 13,7,8$ independent cells from three experiments. **D** Representative recordings from (**C**) of $C_m$ changes (red) conductance (blue) and inward current (pink) after Yoda1 application. All data expressed as mean ± s.e.m. Unpaired Student's $t$-test was used for comparing two groups. Scale bars: 5 μm.

that networks of PM tubules are constricted by dynamins to an extent that they uncouple electrically from the extracellular space. Similar to tight protein–protein interactions in yeast eisosomes and epithelial tight junctions[62], protein–protein contacts between adjacent membranes would prevent ion movements. PL changes mediated by TMEM16F, both cytoplasmic loss of anionic PLs and gain of PC, would cause dynamin relaxation. Importantly, eisosomes are sensitive to complex bilayer properties, including tension and ordering[63,64], and Ca elevations employed here can drastically disorder the PM[65]. All anionic PLs may play important roles, but PIP$_2$ cannot sufficiently do so independent of other anionic PLs[16]. Not only for PM expansion, but also for downstream PM shedding that occurs in Jurkat and HEK cells, the specific PM biophysical changes that are sufficient are only partially defined at this time[17].

Given that channel and transporter activities are minimal in newly exposed PM[16,17], we speculate that compartments might grow by a slow movement of membrane per se through constrictions that close them. In this way, they might remain largely protein-free. In ultrastructural studies, these compartments could be mistaken for PM-associated ER networks[66] or tubular endosomes[67], as observed in both dynamin mutants and knockouts[34,68,69]. In fact, ultrastructural analysis of Dnm double and triple KO MEF cells revealed increased numbers of exposed tubulated pits and fragments[31], consistent with constitutive compartment opening (Fig. 1C and visualized in Supplementary Vid. 4 and Supplementary Vid. 5). Present experiments suggest that Dnm2 plays the major role in PM expansion. Expansion still occurs in Dnm1 KO cells, and Dnm2 expression rescues knockout phenotypes. Dnm2 antibody opens compartments while Dnm1 antibody cannot (Fig. 3D). Impressively, the GTPase-inactive K44A Dnm2 substitutes well for Dnm2 in forming PM reserves and in opening them during PL scrambling (Fig. 1C/D) while the lipid-binding mutant K562E cannot. This underscores the crucial role of lipid binding events and, unlike endocytic events, GTP hydrolysis is not required.

As noted in the Introduction, PM expansion in response to Ca is often greater than available secretory vesicle pools, and we found little evidence for significant organelle involvement (Supplementary Fig. 7). Nevertheless, our previous BHK studies do suggest that PM expansion can occur at low cytoplasmic free Ca via genuine exocytic events, particularly after the occurrence of MEND[16]. Accordingly, cells likely use all three mechanisms to laterally expand their PM, (1) PM unfolding, (2) exocytosis of vesicles and organelles, and (3) opening of dynamin-constricted PM invaginations.

In conclusion, dynamins mediate extensive PM expansion in diverse cells during Ca elevations. Scrambling of anionic PLs, especially PS and PA, initiates expansion and may mediate subsequent inactivation of TMEM16F[17] and PIEZO1[70] as anionic PLs enter the outer monolayer. This form of PM expansion becomes activated in response to increasing PM tension and may play other roles that overlap with those of classical exocytosis.

## Methods

**Statistical analysis and replicate criteria**. Data analysis and figure creation were performed primarily in SigmaPlot v14.5 (Systat) or Matlab v2015 (Mathworks) unless otherwise stated. Excel (Microsoft) was used for general data organization and statistics such as mean, SD, and SEM. Statistical significance was evaluated with Unpaired Students $T$-test for all data sets with normal distributions, For Fig. 2, a Paired Student $T$-test was used compare same cell membrane closure after Trehalose and line scan analysis. The Mann–Whitney Rank Sum Test was employed if the ShapiroWilk normality test failed. Non-specific differences (ns) are defined as $p \geq 0.05$. Outliers were defined as experimental values that deviated more than two standard deviations from the mean in otherwise normally distributed data sets. Outliers were removed only if the number of experiments was >10. Otherwise, the relevant experiments were repeated. All experiments described employed at

least two batches of cells harvested on different days. Data are represented as mean values with SEM. Box plot in Fig. 1 displays median value and upper and lower quartiles with whiskers representing 10$^{th}$/90$^{th}$ percentile. All scale bars are 5 μm unless otherwise noted. Line scan analysis to determine PM depth labeling was performed using 2 separate line scans per cell yielding 4 measurements per cell. The average of the 4 measurements was represented as a single data point in the presentation of PM labeling depth. To estimate the smooth PM area of cells, lengths of the two visible cell axes (i.e., the pseudo-radii) were measured in micrographs. Then, the best estimate of the smooth cell area was calculated from Knud Thomsen's formulas as the average of two possible spheroid surface areas, one with the vertical axis being duplicate in the Z-axis and one with the horizontal axis being duplicate in the Z-axis:

$$A_{Cm} = C_m(A_{Cm} \text{ with } 1\,pF \equiv 110\,\mu m^2) \tag{1}$$

$$A_{smooth} = 4\pi \cdot (((2 \cdot (a \cdot b)^{1.6} + b^{3.2})/3)^{0.625} + ((2 \cdot (a \cdot b)^{1.6} + a^{3.2})/3)^{0.625})/2 \tag{2}$$

Capacitance was converted to surface area[35] and the excess PM (Excess Basal PM or PM$_{excess}$) was then calculated in percent as $100\% \cdot ((A_{Cm}/A_{smooth}) -1)$.

**Reagents**. Unless stated otherwise, reagents were from Sigma-Millipore and were the highest available grade. Ionomycin free acid (Calbiochem) stock solutions were at 5 mM in DMSO and used at 5 μM. TB (Sigma-Millipore) was made fresh at a final concentration of 100 μg/ml (0.01%). Rhodamine-conjugated heptalysine (rhodamine-KKKKKKK-amide; K7) was prepared by Multiple Peptide Systems (NeoMPS, Inc.) and stored at 3 mM stock solutions in H$_2$O with a working concentration of 3 μM. Compared to Annexin V (1:20 dilution from stock, BD Biosciences), K7r was advantageous for real-time imaging as PM binding is significantly faster, and K7r binding does not require Ca. Phosphatidyl-L-serine was from bovine brain (Avanti Polar Lipids, 84002) and L-α-phosphatidylcholine from egg yolk (Avanti Polar Lipids, 840051C). Lipofectamine 3000 (Life Technologies) was used for transient transfection protocols. VAMP2, 4 and 7-pHl constructs were gifts from Prof. Ege Kavalali (Vanderbilt). PIEZO1-ires-GFP construct and Yoda1 were from Ruhma Syeda (UT Southwestern). Dynab nanobody was provided by Aurélien Roux (University of Geneva) and non-myristoylated DIP was purchased from Apex Bio. Antibodies used were from Abcam (Dnm1, Ab52852; Dnm2, Ab3457; BNP, Ab19645). Secondary IgG fluorescent control was goat anti-guinea pig Alex Fluor-488 (Invitrogen, A11073). Antibodies, nanobodies, and DIP were used at a 1:50 dilution in standard KCl buffer (below) without CaCl$_2$ and supplemented with 10 mM EGTA. Control buffers were determined for each antibody per manufacturer and used at 1:50. Fluo-4 AM and x-Rhod-1 AM were purchased from ThermoFisher Scientific and used at 3 μM incubated for 30 min according to manufacturer's protocols. The PA-binding peptide was synthesized by Creative-Peptides (https://www.creative-peptides.com) with the PA-binding sequence FRNEVAVLRKTRHVNILLFMGGYMTKDNLA from Raf-1[50]. Liposomes were prepared by sonication of pure PLs in distilled water, followed by sonication with 3 mM albumin and 10-fold dilution into pipette solutions.

**Buffers employed**. For ionomycin based recordings, cytoplasmic solution contained (in mM): 110 KCl, 5 NaCl, 10 HEPES, 0.5 EGTA, 0.25 CaCl$_2$, and 0.5 MgCl$_2$, adjusted to pH 7.4 and extracellular solution contained (in mM): 120 NaCl, 5 KCl, 10 HEPES, 15 Glucose, 2 CaCl$_2$ and 2 MgCl$_2$, adjusted to pH 7, unless otherwise stated. For NCX recordings, cytoplasmic solution contained (in mM): 80 N-methyl-D-glucamine (NMDG), 40 NaOH, 15 TEAOH, 10 HEPES, 0.5 MgCl$_2$, 0.25 CaCl$_2$, 0.5 EGTA and 120 Aspartic acid adjusted to pH 7.2 with extracellular solutions containing (in mM): 120 NMDG, 10 HEPES, 15 TEA, 0.5 EGTA, 4 MgCl$_2$, and 125 Aspartic acid with 2 CaCl$_2$ added transiently to stimulate reverse NCX activity. Experiments employing intracellular dialysis of buffered calcium employed NCX based solutions with adjustment of EGTA (0–10), pH (6.7–7.2) and CaCl$_2$ to achieve desired free Ca concentrations as calculated by WebMAXChelator (Stanford) and detailed in the text. For Trehalose experiments, NCX buffer was utilized, diluted to 10%, and substituted with 140 mM Sucrose or Trehalose to maintain isosmotic conditions. For cell swelling experimentation, NCX buffer was used in a similar manner as above with 90% substitution of 140 mM sucrose or ddH$_2$O to induce swelling. Additional 2 mM CaCl$_2$ was supplemented to maintain pipette seal and provide Ca influx. Unless stated (i.e., Fig. 3) cytoplasmic buffers all contained physiological Ca buffered with 0.5 EGTA and 0.25 CaCl$_2$.

**Cell culture and cell lines**. Jurkat E6.1T cells (ECACC, cat. 88042803) and H1299 Dnm1 KO cells[32] (modified and provided by S. Schmid, UT Southwestern, ATCC, CRL-5803) were grown in RPMI-1640 (Sigma-Millipore) while BHK-NCX[16] (provided by K. Philipson, UCLA), Inducible MEF Dnm TKO cells[31] (provided by P. DeCamilli, Yale), HEK-293T cells (ATCC, cat. CRL-11268) and Neuro2A cells (provided by E. Kavalali, Vanderbilt, ATCC, CCL-131) were grown in DMEM (Sigma-Millipore). All media were supplemented with 10% FBS (Sigma-Millipore), 2 mM L-glutamine (Sigma-Millipore), 100 U/mL penicillin, and 100 μg/mL streptomycin (Sigma-Millipore). For the H1299 Dnm2 CRISPR KO cells, H1299 cells lacking Dnm1 were treated with the Dnm2-Cas9dd system. Lentiviral constructs,

protocols, and reagents were provided by Prof. S Schmid (UT Southwestern)[33]. In brief, a dilution of 1:100 of viral stock encoding sgRNAs CGATCTGCGGCAG GTCCAGGTGG and CGCCGGCAAGAGCTCGGTGCTGG in the Cas9dd vector was added to fresh media supplemented with 10 μg/mL polybrene (Sigma-Millipore) and added to six-well plates. 50,000 cells/well were added and incubated for 72 h before antibiotic selection with Puromycin (Sigma-Millipore). When cells are incubated with Shield-1, the Cas9dd constitutive degradation system is blocked and subsequent expression of Cas9 allows for the timed deletion of the target sequence, dynamin 2[71]. Shield-1 (Takara) stock concentration was 0.5 mM and used supplemented into fresh cell culture medium at a final concentration of 500 nM 48 h. prior to experimentation. For induction of the dynamin triple KO MEF cell line (Dnm TKO), cells were supplemented with 5 μM InSolution™ Tamoxifen (Sigma-Millipore)[31]. Alternatively, cells were transduced with AAV5-CRE-GFP (Vector Biolabs) in a normal medium for 24 h. Cells were further incubated for 48–72 h. prior to experimentation and selected for Cre expression and induction by the presence of green GFP fluorescence. For WT-Dnm2 rescue, Dnm TKO cells treated with tamoxifen for 5 days were transfected with WT-Dnm2-GFP[10] using Lipofectamine 3000 (Life Technologies). Similarly, K44A Dnm2 rescue utilized Lipofectamine 3000 and cotransfection with mCherry for visualization. Dnm TKO - TMEM16F-null cells were generated using a similar strategy as previously published[17]. Briefly, uninduced Dnm-TKO cells were selected for blasticidin resistance after transduction using lenti-spCas9-Blast (Addgene #52962-LV) according to the manufacturer's guidelines. Resistant cells were further transduced with Sigma Sanger QuickPick™ KO gRNA 3809 and 3810 targeting CTCCAGTGATCCAAAGGTGGGG and TGCCCCACCTTTGGATCACTGG, resp. and selected with puromycin resistance and BFP fluorescence. Both gRNA yielded a similar loss of PM expansion phenotypes with clone 3810 used for data representation. All cell lines are routinely tested for mycoplasma infection and upon arrival using PCR-based detection techniques (ATCC, 30-1012 K). For positive results, Plasmocin™ treatment (InvivoGen, ant-mpt-1) was used to eliminate infection according to the manufacturer's protocol.

**Imaging and patch clamp recordings**. Cells were imaged using the Nikon EZ-C1 confocal microscopy system and patch-clamp recordings of capacitance, current, and conductance were performed on an Axopatch 200B and 1D (Axon Instruments) using Capmeter v7.2 as described before[15,16]. Capmeter code is available at https://sites.google.com/site/capmeter/. In brief, square-wave voltage perturbations (20 mV; 0.5 kHz) were employed during capacitance recordings. Pipette input resistance was 2–9 MΩ with cell seals established at 0.5–2 GΩ. For all imaging and patch-clamp recordings solution or bath chamber temperature was set to 37 °C and solutions were fed via gravity at a flow rate of 2–5 mm/s. A Nikon TE2000-U microscope; ×60 oil immersion, 1.45-NA objective paired with a 40-mW 163-Argon laser (Spectra Physics; Newport Corporation) operating at 488 nm at 5% maximum capacity for pHluorin recordings and a 1.5 mW Melles Griot cylindrical HeNe laser at 543 nm at 55% of maximum capacity for TB and K7r. Emission filters were set to either 500–540 nm (pHluorin) or 580LP for K7r and 630LP for TB. For Trypan Blue PM depth measurements and quantification of K7r binding, fluorescence line scans were determined by subtracting the equivalent background fluorescence. Peak PM binding of K7r fluorescence and PM fluorescence depth in microns was determined from line scan images and quantified in ImageJ using four distinct PM regions per cell. Image analysis was performed using either the EZ-C1 v3.9 (Nikon Instruments) or ImageJ (NIH). When imaging multiple fluorophores, sequential imaging was used to minimize spectral overlap. Photobleaching was negligible for these experiments. For patch clamp, intracellular calcium stimulus was triggered by dialysis of buffered free Ca concentrations ranging from 1 to 7 μM, or by extracellular exposure of ionomycin and Ca for Jurkat, HEK, MEF, N2A, or H1299 cell lines. BHK and HEK cells stably expressing the cardiac Na/Ca exchanger (NCX1.1) as well as isolated primary myocytes were stimulated through reverse transport by cytoplasmic pipette perfusion of cells with 40 mM Na and transient application of 2 mM Ca to the extracellular solution. No difference was detected in ionomycin or NCX-induced stimulation[17].

Additional imaging using a Zeiss LSM 880 Airyscan system paired with a ×63 objective without concurrent patch clamp is described in the corresponding supplemental figure legends. Super-resolution Airyscan mode was utilized with 0.25 μm² Z-stacks starting near the cell adhesion interface were used to visualize dynamin punctae and membrane labeling. Airyscan images were processed using Zeiss Zen software. Punctae loss (Fig. 1) was determined post-processing of thresholded images in ImageJ. AnnexinV-FITC and x-Rhod-1 were imaged using a ×20 objective on a Nikon Eclipse T-2000S equipped with a Clarity (laser free confocal) unit (Aurox) and a Hamamatsu C13440 camera with ex. 466/50; 554/23 and em. 525/45; 609/54 for green and red fluorescence, resp.

**Reporting summary**. Further information on research design is available in the Nature Research Reporting Summary linked to this article.

## Data availability
The authors declare that all data supporting the findings of this study are available from the corresponding authors upon request. The source data within the article, and its Supplementary Data Files, are provided as a Source Data File accompanied with this article. Individual raw data is included on all statistically relevant figures and within the accompanying Source Data File. Source data are provided with this paper.

## Code availability
Capmeter code information is detailed in the Reporting summary linked to this article and available at https://sites.google.com/site/capmeter/.

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

## Acknowledgements

We thank Mei-Jung Lin for technical help and discussion; X. Li (UT Southwestern) for financial and scientific support; P. DeCamilli (Yale) for the inducible triple KO dynamin MEF cell line; S. Schmid (UT Southwestern) for the H1299 Dynamin I KO cell line, dynamin-2-DD-YFP CRISPR lenti, WT-Dnm2-mRuby3, Dnm2-K44A construct and amphiphysin2 SH3 domains; E. Kavalali (Vanderbilt University) for VAMP-pHluorin constructs; J. Albanesi and B. Barylko (UT Southwestern) for WT-Dnm2-GFP, Dnm2-K44A-GFP and Dnm2-K562E-GFP; and Aurélien Roux (University of Geneva) for Dynab nanobodies. We express a special thank you to Dr. Orson Moe for his invaluable contributions, discussions, and support. DH and MF were supported by the National Institutes of Health, USA (HL119843, T32DK007257). R.S. was supported by UT Southwestern Medical Center Endowed Scholar Program and American Heart Association grant (17SDG33410184).

## Author contributions

M.F., D.H., and C.D. carried out electrophysiological experiments. M.F. contributed optical acquisition and analysis. R.S. provided PIEZO constructs and reagents as well as assisted in mechanosensitive analysis and preparation of the article. All authors contributed to data analysis and contributed to manuscript preparation. M.F. and D.H. conceived the project and wrote the manuscript.

## Competing interests

The authors declare no competing interests.
