## [Peer Review File · Nature Communications]

REVIEWER COMMENTS

Reviewer #1 (Remarks to the Author):

The paper by Fine et al follows on the previous work published in 2018 by the Hilgemann lab. That study examined the function of TMEM16F, an ion channel and Ca dependent lipid scramblase. The conclusion of the previous work was that continued ionophore application in Jurkat T cells caused extensive shedding of ectosomes containing the T cell co-receptor PD-1. Cells lacking TMEM16F, when stimulated by high calcium, underwent the opposite response, massive endocytosis (MEND) -- loss of plasma membrane surface area. In the current paper, the authors show that Ca activation of TMEM16F opens invaginations in cells to increase membrane area, and again show MEND in the absence of the scramblase.

The present work repeats many of the conclusions established in Jurkats and extends them to multiple types of cell lines. The key difference in this paper is that the authors argue that the most physiologically relevant action is that calcium increase (e.g., from mechanical stress/Piezo) induces loss of dynamin (Dyn2 primarily) necks that normally constrict and localize plasma membrane invaginations, thus releasing this 'new' PM to the surface. This expansion allows the cells to adapt to changing conditions, especially when their borders may be constrained by tight junctions.

Overall the work is excellent in that it provides an objective and highly quantifiable measurement, change in membrane capacitance, to measure the time course and absolute change in plasma membrane surface area. The authors use orthogonal methods and tools to verify that dynamin neck release and scramblase flipping of phosphatidylserines enables this transition.

Major

1. The authors should place in context their previous conclusions about exosomes in light of the current data. Is exosome release vs PM expansion a product of level of Ca increase? I have the feeling that the authors think exosome release is an extreme response to more severe conditions, but this is not made clear. Is exosome release more common in unbounded single cells like T cells? What happens in tightly constrained epithelial tissues? I am not requesting new experiments, just more discussion of these issues.
2. The one point that could be improved in the current ms is the imaging of the in incursions into cells labeled by Trypan Blue. The videos help, but certainly EM, or at least higher resolution, would improve understanding of the nature of these incursions. Since during this pandemic, such experiments may be impractical, I suggest these data be added only if possible.

Reviewer #2 (Remarks to the Author):

This paper proposes a novel mechanism for membrane expansion that involves lipid scrambling and opening of cytoplasmic membrane compartments that are isolated from the plasma membrane by dynamin constrictions. While the data are very interesting, I think that the paper is undercooked and requires more work to be convincing.

- (1) It seems that the paper consists of two incomplete stories: 1) The effects of dynamin on the

changes in capacitance and 2) The effects of piezo1 on capacitance. In my view these stories are not connected well and have missing pieces. The connection of piezo1 and dynamin is not strong (there is none) and does not warrant the model in Fig 4D. My suggestion would be to focus on the dynamin angle, which is the strongest.

(2) The proposal that the P_{Ser} scramblase activity of TMEM16F is the first step to release a dynamin membrane constriction is not supported by the data and it does not agree with the known fact that PI_{4,5}P lipids are the main way dynamins are held on membranes. To my knowledge, it is not known whether PI_{4,5}P is a substrate for TMEM16F (or any scramblase in the TMEM16 family). I would suggest testing the participation of PI lipids using tools such as recombinant synaptojanin perfusion or rapid Sac1 delivery to the plasma membrane PMID: 22722250. One could also test the effect of the voltage-activated phosphatase.

(3) The results in Fig 1D-E can be interpreted either as TMEM16F and dynamin are in the same pathway or they are in parallel pathways. There is no way to discern between these models. Additional data is required. For example, the authors should consider rescuing the effect of TMEM16F gain-of-function mutants by dynamin mutants or KO.

(4) Because the expected phenotype from a dynamin knockout is not immediately obvious, the interpretation of the results seems forced. If dynamin normally constricts the neck of the cytoplasmic membrane compartment, what would one expect to happen to this compartment in the absence of dynamin? It seems there are several possibilities. One is that the internal compartments would form normally but would be constitutively continuous (electrically) with the plasma membrane. Are the Dnm TKO cells larger than WT? If not, why not? Other possibilities could be related to completely messed up membrane trafficking. Does MEND occur in the Dnm TKO (induced) -TMEM16F null, as one would expect? The authors should investigate the effects of these mutations in more detail.

(5) Statements about pits on the membrane that are closed or open (model in Figure 4) need better evidence. The microscopy presented is insufficient. For example, in Movie 1 it seems that between 6-9 minutes there are K7r positive puncta that appear and disappear. What is going on? Movies 2 and 3 – hard to tell whether these are omega figures or endocytosis.

(6) The authors should contrast the findings and compare with eisosomes in yeast.

(7) Several experiments lack appropriate controls. For example, Amphiphysin SH3 domains needs a control. There are no positive controls for SFig. 4 to show that the methodology works. Fig. 3B needs a control of 10mM EGTA + AMPPNP without protamine.

(8) The authors use trypan blue as a probe and cite ref 13, but I cannot find any use of trypan blue in this publication. Their claims about its properties need verification. On line 141, it is stated that TB fluoresces red when bound to PM and cites ref 34, but this reference is about yolk granules.

(9) The authors conclude that loss of anionic lipids from the cytoplasmic monolayer is a trigger for opening the cytoplasmic compartments. This seems like a jump. If the idea is that these polycations work by masking the negative surface charge, wouldn't the elevation of cytosolic Ca alone be expected to mask the negative charge as well (without the need for scrambling)?

Reviewer #3 (Remarks to the Author):

The current manuscript investigates the mechanisms underlying plasma membrane (PM) expansion, a process that allows cells to cope with mechanical perturbations and mediate processes such as phagocytosis. Recent work by the authors' group showed that activation of the TMEM16F scramblase can trigger PM expansion. Here, the authors show that this process entails the release of membrane compartments that are isolated from the PM by dynamin-delimited constrictions. Depletion of PS from the inner leaflet, via activation of the TMEM16F scramblase, internal dialysis of polycationic peptides or of dynamin binding peptides, induces opening of these compartments. The authors show that Ca²⁺ influx via the PIEZO1 channel can activate the TMEM16F scramblase to initiate this process.

While the topic of the manuscript is interesting and the topic of broad importance, I have several major concerns that need to be addressed. Additionally, I found the writing hard to follow. The descriptions of the experiments and results could be expanded to allow for readers to better follow the presentation.

- While the data shows that dynamin and TMEM16F are involved in the PM expansion process, the proposed mechanism is not well supported. Several key pieces of evidence are missing:

- i. The authors need to show that dynamins and the TMEM16F scramblase specifically localize at the neck of the invaginations whose opening underlies the reported PM expansion.
- ii. It is also important to provide direct evidence of the specific co-localization of these two proteins at the neck of these invaginations.
- iii. The authors should show that dynamin localization is altered by TMEM16F activation: at rest dynamin localizes at the neck of these invaginations and that upon activation of the scramblase dynamin moves away.

These experiments are critical to validate and support the proposed mechanism for the involvement of these proteins in PM expansion.

- It is unclear how PS scrambling specifically impairs dynamin binding to the neck of these invaginations. PS is presumably present throughout the inner leaflet of the PM, and TMEM16F activation results in PS externalization throughout the cell. If PS is present throughout the inner leaflet, how does dynamin 'know' where to specifically cluster and organize to form the tight constrictions necessary to isolate these invaginations? Are these compartments particularly enriched in PS? If so, this should be experimentally demonstrated. If not, then what drives dynamin to form these constrictions?

- Are cells lacking TMEM16F (and/or dynamin) more susceptible to hypotonic stress? The authors' conclusions on the role of these proteins in PM expansion process would predict that the response of cells lacking these proteins to hypotonic stress should be impaired. This should be tested.

- The authors claim to identify the PM compartment that is opened by upon activation of TMEM16F and dynamins as the enlargeosomes. However, this claim is not well founded in the results. A more direct comparison between the properties of enlargeosomes and the compartments identified here is needed to draw the identification drawn by the authors. The evidence that VAMP4-containing compartments are preferentially fused into the PM is not sufficient to draw firm conclusions as this

marker is not only present in enlargeosomes. Rather, VAMP4 is associated with the trans-Golgi network and in early endosomes, which argues against the authors' conclusion that PM expansion is not mediated by endosome fusion.

- The authors' data suggests that Ca^{2+} entry through PIEZO1 can play a role in activating TMEM16F. However, the experiments described here were performed under overexpression conditions casting serious doubts on whether this occurs in physiological conditions, as claimed by the authors, or if it is due to the artificial experimental set up. The authors should repeat these experiments in cells that endogenously express both PIEZO1 and TMEM16F.

- I am puzzled by the authors' inclusion of PIEZO1 among the channels that are activated by cell swelling. Neither of the reviews associated with this assertion contain data (or references to) that support this notion. Rather, hypotonic stress activates the LLRC/SWELL channels, which are anion selective and Ca^{2+} impermeable. The authors should rule out the possibility that the effects seen here are mediated by these channels, i.e. showing that the responses seen here are not affected by LLRC-specific inhibitors.

- Is the capacitance of cells lacking dynamin and/or TMEM16F constitutively higher than that of WT cells? If not, what happens to the excess membrane? Is it not synthesized?

RESPONSE TO THE REVIEWS.**TMEM16F and dynamins control expansive plasma membrane reservoirs.**

We thank the reviewers for their helpful comments and some astute criticisms which have guided our extensive revision of this article. At a scientific level, our major improvements are the following:

- (1) We provide new data supporting dynamin involvement in membrane expansion. These include descriptions of compartment opening with dynamin-inhibitory peptides in multiple cell types.
- (2) We provide new experiments demonstrating that dominant-negative (GTPase-inactive) dynamin2 can substitute for dynamin2 in its roles to form sequestered plasma membrane compartments and to open them when phospholipids scramble. These experiments document that the phenotypes of cells with dynamin2 deletions and with expression of dominant negative dynamins are drastically different. Beyond reasonable doubt, our working hypothesis goes far to explain these fundamental results.
- (3) We provide clear evidence that the expansion process is used by cells to extend their plasma membranes during cell swelling, and that phospholipid scrambling occurs in parallel with this response. Therewith at least one physiological role of the expansion process is firmly established.
- (4) We provide new evidence that compartment opening depends on phospholipid scramblase activity, not just a loss of anionic phospholipids but also a movement of phosphatidylcholine in the inner monolayer.

At the editorial level, our major improvements include the following:

- (1) We have improved the writing so as to lead the reviewer through our data in a more clear and organized fashion, starting with mechanistic data and ending with physiology-relevant data
- (2) We have moved data about expansion with low free Ca to the Supplement so as to allow presentation of new data about dynamin mutants and the roles of neutral phospholipids.
- (3) We have strived to create a text that will be attractive to readers with a wide range of interests from immunology to neuroscience, cell biology, and membrane biophysics.

We provide the following point-by-point response to the reviewers' comments:

Reviewer #1 (Remarks to the Author):

The paper by Fine et al follows on the previous work published in 2018 by the Hilgemann lab. That study examined the function of TMEM16F, an ion channel and Ca dependent lipid scramblase. The conclusion of the previous work was that continued ionophore application in Jurkat T cells caused extensive shedding of ectosomes containing the T cell co-receptor PD-1. Cells lacking TMEM16F, when stimulated by high calcium, underwent the opposite response, massive endocytosis (MEND) -- loss of plasma membrane surface area. In the current paper, the authors show that Ca activation of TMEM16F opens invaginations in cells to increase membrane area, and again show MEND in the absence of the scramblase.

The present work repeats many of the conclusions established in Jurkats and extends them to multiple types of cell lines. The key difference in this paper is that the authors argue that the most physiologically relevant action is that calcium increase (e.g., from mechanical stress/Piezo) induces loss of dynamin (Dyn2 primarily) necks that normally constrict and localize plasma membrane invaginations, thus releasing this 'new' PM to the surface. This expansion allows the cells to adapt to changing conditions, especially when their borders may be constrained by tight junctions.

Overall the work is excellent in that it provides an objective and highly quantifiable measurement, change in membrane capacitance, to measure the time course and absolute change in plasma membrane surface area. The authors use orthogonal methods and tools to verify that dynamin neck release and scramblase flipping of phosphatidylserines enables this transition.

Major

1. The authors should place in context their previous conclusions about exosomes in light of the current data. Is exosome release vs PM expansion a product of level of Ca increase? I have the feeling that the authors think exosome release is an extreme response to more severe conditions, but this is not made clear. Is exosome release more common in unbounded single cells like T cells? What happens in tightly constrained epithelial tissues? I am not requesting new experiments, just more discussion of these issues.

We thank the reviewer for helpful as well as supportive comments. We agree that it is important to highlight membrane shedding. Shedding is coupled to the expansion process, as demonstrated in our previous report in the Jurkat cell line, as well as to some extent in HEK and BHK cells. (Bricogne, et. al., *Scientific Reports*, 2019). We now bring this out both in the Introduction and the Discussion. We point out that the shedding process is a relatively cell-dependent process. In response to rapid and prolonged increases in Ca, Jurkat cells shed significantly while HEK and BHK cells do so to a lesser extent. In MEF cells and cardiomyocytes, shedding is virtually absent. In all probability, the shedding is indeed coupled to phospholipid scrambling. We have added Supplemental Video 1 revealing PM scrambling and Ca influx in Jurkat cells after prolonged ionomycin/Ca giving rise to subsequent membrane shedding. We have now also provided a representative image, similar to Supp. Video 1 showing shedding with FM 4-64 dye as well (Sup Fig. 1E). Given space limitations, we cannot expand on this subject further. In summary, the reviewer is correct that exosome release is more prominent in cells like T cells and is likely associated with cell specific responses, and this represents a significant avenue for further work.

The one point that could be improved in the current MS is the imaging of the in incursions into cells labeled by Trypan Blue. The videos help, but certainly EM, or at least higher resolution, would improve understanding of the nature of these incursions. Since during this pandemic, such experiments may be impractical, I suggest these data be added only if possible.

We agree that additional imaging and especially ultrastructural approaches will be needed in the long term. New EM etching techniques are now becoming available to generate 3D structures of all membranous structures in cells. While we have initiated collaborations in this area with positive preliminary data, it is beyond our present scope to provide such analyses.

For now, therefore, we have improved by including better images with our present methods, and we provide new supplementary videos with clearer labeling of the newly expanded compartments. Some images achieve effectively super-resolution for light microscopy, and they reveal well, within these limits, the structures of the membrane compartments that open.

Importantly, EM studies of the Dnm DKO/TKO MEF cells are long available from the De Camilli group (Park, R. et. al., *JCS*, 2013; Ferguson S. et. al., *Dev. Cell*, 2010). These studies reveal an impressive network of new plasma membrane-coupled compartments in the knockout cells. They were described as “tubulated pits” and “tubulated fragments”, and they were assumed in first order to represent 'frustrated' endocytic events. However, they may instead, or in addition, represent the compartments involved in membrane expansion. Accordingly, we include more detailed but careful comments about these studies in the new Discussion.

Reviewer #2 (Remarks to the Author):

This paper proposes a novel mechanism for membrane expansion that involves lipid scrambling and opening of cytoplasmic membrane compartments that are isolated from the plasma membrane by dynamin constrictions. While the data are very interesting, I think that the paper is undercooked and requires more work to be convincing.

(1) It seems that the paper consists of two incomplete stories: 1) The effects of dynamin on the changes in capacitance and 2) The effects of piezo1 on capacitance. In my view these stories are not connected well and have missing pieces. The connection of piezo1 and dynamin is not strong (there is none) and does not warrant the model in Fig 4D. My suggestion would be to focus on the dynamin angle, which is the strongest.

We now show that cell swelling indeed activates the TMEM16F pathway with new data in Figure 4 (Fig. 4B). Specifically, we demonstrate that cell swelling by hypotonic solutions generates Ca influx-dependent phospholipid scrambling that requires TMEM16F and extracellular Ca, and that leads to plasma membrane expansion. We also now include data ruling out TMEM16F as the conduction pathway for the Ca signal, indicating that additional channels or pathways must be involved. The original experiments (now Fig. 4C/D) employing 'expressed' PIEZO channels provided an example of a Ca-influx pathway that could mediate membrane expansion, and we acknowledge that our work does not implicate PIEZO channels per se. We have therefore removed the naming of PIEZO specifically from the model. We simply indicate graphically that Ca influx occurs, independent of TMEM16F, and that the Ca influx activates TMEM16F scrambling followed by membrane expansion during cell swelling (and likely other mechanical perturbations). We have also improved the discussion in this context by pointing out previous work on Ca-dependent membrane expansion during cell swelling by the Cahalan group in Jurkat cells (Ross and Cahalan, *JGP*, 1995). Recent communication with Prof. Cahalan regarding the latest developments of these swelling associated cation channels (SWAC) further supports our observations and will lead to future collaborations and reports with a focus on identifying possible SWAC candidates involved in swelling mediated Ca-entry. We try now simply to underscore the importance of these processes for cells while point out that the specific ion channels involved remain to be ascertained.

(2) The proposal that the Pser scramblase activity of TMEM16F is the first step to release a dynamin membrane constriction is not supported by the data and it does not agree with the known fact that PI4,5P₂ lipids are the main way dynamins are held on membranes. To my knowledge, it is not known whether PI4,5P is a substrate for TMEM16F (or any scramblase in the TMEM16 family). I would suggest testing the participation of PI lipids using tools such as recombinant synaptojanin perfusion or rapid Sac1 delivery to the plasma membrane PMID: 22722250. One could also test the effect of the voltage-activated phosphatase.

Numerous articles over decades demonstrate functionally important dynamin interactions with anionic lipids other than phosphoinositides. These lipid species include both phosphatidylserine and phosphatidic acid, both of which definitively scramble in a Ca-dependent manner. In fact, phospholipids with very large head groups have been known for a long time to scramble in response to high Ca in RBCs (e.g., Dekkers et al., PMID: 11879203). More recently, it has been demonstrated that

TMEM16F can mediate scrambling of phospholipids with very large head groups (Malvezzi et al., PNAS, PMID:29925604). Thus, it must be expected that PIP2 as well as PA and PS undergo scrambling.

We have included new references in the revised manuscript concerning the roles of multiple anionic lipids in the regulation and function of dynamins. We agree with the reviewer that this is a topic of *significant* importance as highlighted by recent work from the lab of Lily Jan, and we are careful in the present article to NOT downplay a possible contributing role of PI4,5P₂. In fact, we have previously reported a small but significant (< 8%) PM expansion response during PLC-mediated cleavage of PI4,5P₂. (Yaradanakul et. al., *JGP* 2008; Wang and Hilgemann, *JGP* 2008). In those experiments we transiently activated PLC through overexpression of hM1 receptors and application of the agonist carbachol. As suggested by the reviewer, we have reiterated our previous findings regarding phosphoinositide cleavage and PM expansion. In addition, we have improved the perspective/discussion regarding this topic. Recently, we performed preliminary experiments using the dr-VSP voltage sensitive phosphatase in so far as monitoring voltage induced dephosphorylation of PI4,5P₂ using a fluorescent pleckstrin homology domain (PH). PM expansion during these pulses is minimal compared to the extent by which PS and PA or TMEM16F mediated scrambling can induce. For now, this data is too preliminary for inclusion within this manuscript especially in combination with previously published findings reiterating similar observations.

More importantly, due in no small part to the advice of the reviewer, we describe in the revised article the importance on additional lipids involved in scrambling and PM expansion that have significantly broadened the scope of our research. We now demonstrate that an increase of phosphatidylcholine in the inner monolayer can support membrane expansion and can synergize with chelation of anionic phospholipids to support membrane expansion in Jurkat cells. These new results make a strong case that the membrane changes that activate membrane expansion are more complex than simply a loss of anionic phospholipids from the inner monolayer. While dynamin lipid interactions are driven primarily by anionic lipids and the most abundant of these is PS, clearly the contribution of neutral lipids like PC reveal more complex biophysical principles, likely involving changes of membrane ordering as well as local lateral tensions. To be clear, we now show that scrambling, defined as the movement of lipids in either direction along their transmembrane concentration gradients, promotes the membrane expansion responses described, likely by causing a relaxation of dynamins. Local interactions of dynamins with TMEM16F are not required, although the decline of TMEM16F activity that occurs as membrane expansion slows is highly consistent cytoplasmic anionic lipids being activators of TMEM16F. In this connection, the Lily Jan group has recently published that PI4,5P₂ directly inactivates TMEM16F channels (W. Ye, et. al., *PNAS*, 2018). We believe that other anionic lipids are also important, both for TMEM16F and for dynamin function. In the case of membrane expansion, and thereby likely dynamin relaxation, membrane changes besides loss of anionic phospholipids clearly play important roles to regulate PM compartment opening. This is highlighted in the revised Fig. 3 as well as Supplemental Files, and it would not be surprising if TMEM16F regulation lipids turned out to be similarly complex.

(3) The results in Fig 1D-E can be interpreted either as TMEM16F and dynamin are in the same pathway or they are in parallel pathways. There is no way to discern between these models. Additional data is required. For example, the authors should consider rescuing the effect of TMEM16F gain-of-function mutants by dynamin mutants or KO.

We now show that dominant negative dynamin2, namely K44A (Song et al., JBC, PMID15262989), substitutes well for WT dynamin2 to both generate sequestered membrane compartments to allow them to open. For the first time, we show beyond any possible doubt that GTPase-inactive dynamin2 has important biological functions that are opposite from the effects of deleting dynamins. We point out further in response to this comment that we have demonstrated in Fig. 1D the rescue of the TMEM16F-null phenotype by expression of mTMEM16F as well as several TMEM16F mutations, as detailed in our previous publication (Bricogne C. et. al., *Scientific Reports*, 2019). In the revised article, we have improved the description of these results within both the text body and legends. We underscore our previous results for TMEM16F mutants, indicating that PM expansion requires the phospholipid scrambling function of TMEM16F. In that study, we reported rescue of TMEM16F-null cells with a constitutively active mutant, D430G, and an ion selectivity/activation mutant, Q559K. Both mutants allow PM expansion in response to ionomycin, although the Q559K mutant has reduced scrambling activity and promotes a slower expansion response.

Performing these experiments with dynamin mutants or KO cell lines should not alter the pathway, since dynamin is well demonstrated by our present experiments to regulate PM expansion downstream of TMEM16F-induced scrambling. It is expected therefore that the phenotypes would be comparable to the outcomes described in this manuscript. While the suggested experiments are interesting, we strongly believe that our demonstration of Dnm knockout rescue by GTPase-inactive dynamin2 is far more important than the proposed experiments for the present article. We also clarified this concept in Figure 1E where we describe in more detail how scrambling is preserved in induced Dnm TKO cells where dynamin is missing, however PM expansion is blocked. Likewise, Dnm TKO cells lacking TMEM16F fail to scramble and do not expand indicating PM expansion is subsequent to PL scrambling.

Together, the combination of KO, DKO, TKO and rescue experiments all support the involvement of both TMEM16F and dynamin in the Ca-dependent membrane expansion pathway. Additional data directly manipulating the anionic lipid availability to dynamin (through sequestration) further supports the pathway delineated by us. Our further new data demonstrates that Ca entry via channels activated by osmotic stress occurs independent of TMEM16F, but requires TMEM16F to cause membrane expansion. In addition we have included data, similar to our previous observations (Bricogne C. et. al., *Scientific Reports*, 2019), using the PS specific probe Annexin V to detect PS-specific scrambling during TMEM16F activation and membrane expansion (Fig. 4), as well as during cell swelling that leads to 'physiological' activation of (probably local) lipid scrambling and PM expansion.

(4) Because the expected phenotype from a dynamin knockout is not immediately obvious, the interpretation of the results seems forced. If dynamin normally constricts the neck of the cytoplasmic membrane compartment, what would one expect to happen to this compartment in the absence of dynamin? It seems there are several possibilities. One is that the internal compartments would form normally but would be constitutively continuous (electrically) with the plasma membrane. Are the Dnm TKO cells larger than WT? If not, why not? Other possibilities could be related to completely messed up membrane trafficking. Does MEND occur in the Dnm TKO (induced) -TMEM16F null, as one would expect? The authors should investigate the effects of these mutations in more detail.

We agree that the expected phenotypes for Dnm TKO cells were not immediately obvious at first, nearly independent of the current working hypotheses of how dynamin functions in canonical endocytosis.

However, our new work shed much light on this problem, namely by demonstrating that GTPase-inactive dynamin2 can alleviate many of the gross phenotypes of the triple dynamin knockout. YES, there are still many possible outcomes and complexities. But we show in Figure 1B that (1) PM excess area is increased constitutively in the Dnm TKO cells, (2) this vast increase is reversed by dominant negative dynamin2, and (3) dominant negative dynamin2 strongly reduces the excess PM area in WT HEK293 cells. With good certainty, the excess membrane area reflects the presence of membrane infolding that is continuous (electrically) with the PM, while the loss of excess area reflects mostly the sealing off of PM by dynamins. In addition, we point out that in Dnm TKO cells with TMEM16F-null, MEND can indeed occur (See Sup. Fig. 1, and subsequent paragraph). However interesting the related topics of MEND and PM expansion may be, they go well beyond the focus of this manuscript and will likely require several additional studies to clarify the relationship. For now, the focus of this manuscript is on the relationship between scrambling, PM expansion, dynamin and the potential physiological sources for Ca-entry that lead to such phenotypes.

Furthermore with regard to MEND in the TKO cells, we point out that we have published a review article demonstrating that Ca can activate massive endocytosis (MEND) in these Dnm TKO cells (Hilgemann et al., BBA, PMID31202864). We have noted this observation specifically in the text and in Supplemental Figure 1 (Yellow trace). Additionally, electron micrographs of Dnm DKO and TKO MEF cells from the lab of Pietro De Camilli (as described for Reviewer #1) are at minimum consistent with the notion that the internal compartments can remain relatively intact and open to the extracellular space after deletion of dynamins. Given the additional new data provided in the revised manuscript, and improved discussion regarding previous work, we feel that further mutational analysis would exceed a reasonable scope, as well as the space limits, of the present article.

(5) Statements about pits on the membrane that are closed or open (model in Figure 4) need better evidence. The microscopy presented is insufficient. For example, in Movie 1 it seems that between 6-9 minutes there are K7r positive puncta that appear and disappear. What is going on? Movies 2 and 3 – hard to tell whether these are omega figures or endocytosis.

We have enhanced the description of the supplemental images and videos to underscore the punctae formed during membrane expansion remain intact for substantial periods of time when cells are NOT hypo-osmotically swollen. As emphasized in the revised supplement, these punctae disappear when the extracellular dye is removed, thus indicating that this cannot be endocytosis. It is simply the washoff of the extracellular dye. To specifically address the time points mentioned (6-9 min) we point out that these experiments are performed with dialysis of 5 uM free Ca and that scrambling and PM expansion occurs slower in this context (See Fig. 1 and Supp. Fig 1). As TMEM16F initiates PS exposure, the external K7r dye binds to that location. In the beginning, this occurs as small localized punctae as noted by the reviewer, over time the punctae remain exposed but anionic lipids may distribute laterally causing diminished K7r binding to these discrete spots. As TMEM16F activation continue, anionic exposure occurs with more punctae opening, revealing increased K7r binding that eventually spreads across the entire bilayer as observed toward the end of the video. This is also demonstrated in Supplemental Figure 2 showing time course and line traces of K7r. The reviewer is correct that less-specific probes such as Trypan Blue or FM dyes, instead of K7r, would label the compartments more consistently as they are less specific for anionic lipids (See the new Supplemental Video 4). In addition to improving the description and images indicating labeled compartments exposed to the extracellular space, we point out again the work of the De Camilli group, describing the presence of increased tubulated pits and

fragments in the Dnm DKO and TKO cells (Park, R. et. al., *JCS*, 2013, PMID: 24046449 ; Ferguson S. et. al., *Dev. Cell*, 2010, PMID: 21169560). In those experiments, only compartments with exposed opening to the outside environment would be labeled and detected during EM analysis. To address whether the labeled Omega figures in Supp. Video 2,3 are bona fide endocytosis, we point out that these cells are fixed without permeabilization and labeled externally with Trypan Blue (a non-permeable dye) after fixation. In this context, if the cells were permeable the label would saturate the cytosol and nucleus with fluorescence binding. The labeling, post-fixation, in this high-resolution imaging Z-stack clearly reveals only exposed surface labeling of the cells and clearly excludes labeling of intracellular compartments. We must conclude that the clear increase in tubules and Omega figures is indicative of invaginated compartments, contiguous with the cell surface and open to the extracellular space. These compartments are clearly much less pronounced in control cells, not treated with ionomycin and Ca.

(6) The authors should contrast the findings and compare with eisosomes in yeast.

We agree that eisosomes are of clear relevance to the present article and might represent a mechanistically related membrane compartment in yeast. Anionic phospholipid (PI4,5P₂) sensitivity and BAR domain interactions are certainly enticing. Within the present article, we now briefly mention this and reference the most obviously relevant work in the Discussion.

(7) Several experiments lack appropriate controls. For example, Amphiphysin SH3 domains needs a control. There are no positive controls for SFig. 4 to show that the methodology works. Fig. 3B needs a control of 10mM EGTA + AMPPNP without protamine.

Regarding controls for the SH3 domains, we can state that multiple domains and peptides do not have equivalent effects (See revised Fig. 1 and Fig. 3). Buffer controls are more clearly pointed out within the text and figure legends. We have now also included new data on the SH3 domain in multiple cell lines (See revised Fig 1), as well as the dynamin inhibitory peptide (DIP, Fig. 3D). The amphiphysin SH3 domain binds the PRD domain of dynamin while the DIP is based on the fragment of the dynamin PRD domain that binds amphiphysin. We have included a significant revision with multiple new datasets for Fig. 3 to address the reviewer's concerns, lending further support to the role of dynamin-lipid interactions in membrane expansion. We now provide a data set in Fig. 3C showing that in our conditions to analyze PM expansion, the non-hydrolyzable ATP analogues, AMPPNP, does not have a significant effect in Jurkat cells. However, we did notice during our revision work that nonhydrolyzable ATP analogues can promote a relatively minimal PM expansion responses in some cell types under our experimental conditions. To simplify, the existing expanded data set (unless otherwise noted) no longer contains non-hydrolyzable ATP analogues and buffer controls are Ca-free with EDTA or EGTA as noted. After testing several interventions that might block ATPase activities, we settled on 10 mM EDTA to chelate all Mg in cells and thereby block ATP hydrolyzing processes. As shown now in Fig. 3C, EDTA causes a significant but small expansion response, and this response is fully blocked by orthovanadate. While we cannot prove that this block is caused by inhibition of lipid phosphatases at this time, we are very suspicious that this is the case. First, it is known (as indicated in our revision with a reference) that the PM of epithelial cells contains a powerful phosphatidate hydrolase that is highly sensitive to vanadate block and is completely insensitive to Mg chelation or enhancement. The second reason, we are increasingly suspicious about a role for phosphatidate is that we can induce substantial PM expansion in cells by simply dialyzing cells with at specific PA binding peptide. The concentration that we

are employing (10 μ M) is in a range that is suggested to be specific for PA versus other anionic phospholipids.

Further, with regards to controls for Sup. Fig 4, we have included additional data, description and images/videos to support our conclusions. We point out that the data set regarding VAMP proteins is effectively a control for our studies of organellar fusion. In other words, our methods are effective in identifying proteins in the compartments that open. In the data set shown in Sup. Fig. 5 and Sup. Vid. 7 the luminal pH tag increases exposure after PM expansion for Vamp4, but not Vamp2 and to a much lesser extent than Vamp7. These crucial differences give additional support for our protocols that the quenching of luminal fluorescent probes after PM expansion using both TB and colocalization analysis can yield significant data.

(8) The authors use trypan blue as a probe and cite ref 13, but I cannot find any use of trypan blue in this publication. Their claims about its properties need verification. On line 141, it is stated that TB fluoresces red when bound to PM and cites ref 34, but this reference is about yolk granules.

The reviewer is correct. Our TB references were spoiled at some point, and we did not notice. Correct refs. are now in place (Busetto et. al., and Harrisson et. al.). That said, the citation given (originally 34), regarding yolk granules, is a valid description of the fluorescence excitation and emission profile of Trypan when bound to protein (See. Fig. 2).

Further to Trypan Blue fluorescence, Fig. 2A reveals a tight correlation between membrane capacitance and TB fluorescence changes with a Pearson correlation coefficient of 0.9 and slope, 0.96 ± 0.03 . These results indicate that, as employed here, the cellular membrane fluorescence of TB, applied to the outside of cells, correlates much better to cell surface area than does the fluorescence of other dyes employed in related studies.

(9) The authors conclude that loss of anionic lipids from the cytoplasmic monolayer is a trigger for opening the cytoplasmic compartments. This seems like a jump. If the idea is that these polycations work by masking the negative surface charge, wouldn't the elevation of cytosolic Ca alone be expected to mask the negative charge as well (without the need for scrambling)?

This is a good point, as it may indeed be the case that Ca reaches high enough concentrations in some experiments to bind to anionic phospholipids. The situation is that one of the fast forms of massive endocytosis occurs when TMEM16F is not present. Thus, one possibility is that Ca binding to anionic phospholipids tends to both promote both membrane expansion and to cause MEND. The scrambling of phospholipids however prevents the occurrence of such MEND by taking many anionic phospholipids out of the inner monolayer. The fact is that the rate of expansion follows precisely the TMEM16F conductance time course (Bricogne et. al., *Sci. Reports*, 2019) and the appearance of anionic phospholipids in the outer monolayer (See Supp Fig. 2/3). Most importantly, we describe in the new article that loss of cytoplasmic anionic phospholipids is one of but with certainly not the only trigger for membrane expansion. Expansion in response to polycations remains substantially smaller than expansion with Ca elevations, and we demonstrate that an increase of cytoplasmic phosphatidylcholine also promotes expansion. In short, the membrane changes occurring are with certainly highly complex, and it seems certain now that multiple effects synergize to trigger the full expansion responses observed with Ca elevation.

Reviewer #3 (Remarks to the Author):

The current manuscript investigates the mechanisms underlying plasma membrane (PM) expansion, a process that allows cells to cope with mechanical perturbations and mediate processes such as phagocytosis. Recent work by the authors' group showed that activation of the TMEM16F scramblase can trigger PM expansion. Here, the authors show that this process entails the release of membrane compartments that are isolated from the PM by dynamin-delimited constrictions. Depletion of PS from the inner leaflet, via activation of the TMEM16F scramblase, internal dialysis of polycationic peptides or of dynamin binding peptides, induces opening of these compartments. The authors show that Ca²⁺ influx via the PIEZO1 channel can activate the TMEM16F scramblase to initiate this process.

While the topic of the manuscript is interesting and the topic of broad importance, I have several major concerns that need to be addressed. Additionally, I found the writing hard to follow. The descriptions of the experiments and results could be expanded to allow for readers to better follow the presentation.

We have worked diligently to improve our messages and focus. We have improved the descriptions of experiments and results as described above and below, so readers should be able to follow the presentation much more easily.

- While the data shows that dynamin and TMEM16F are involved in the PM expansion process, the proposed mechanism is not well supported. Several key pieces of evidence are missing:

i. The authors need to show that dynamins and the TMEM16F scramblase specifically localize at the neck of the invaginations whose opening underlies the reported PM expansion.

Clearly we did not adequately explain our hypothesis. We now explicitly point out that our hypothesis DOES NOT require co-localization of dynamins and TMEM16F. Dynamin is an adaptor bound to the membrane, while TMEM16F is also an integral membrane protein. Rates of lateral diffusion of lipids in the membrane do not require that the proteins co-localize to allow the activation of membrane expansion. Certainly, it would be of great interest if this were the case, but it need not be. In this connection, we stress now that our findings do indeed support that TMEM16F co-localizes with a mechanosensitive Ca-permeable channel involved in membrane expansion during cell swelling. As described, this can occur with expressed PIEZO1 channels, although other mechanosensitive channels may mediate the expansion that occurs with cell swelling. We have clarified these points in the text body and Discussion.

ii. It is also important to provide direct evidence of the specific co-localization of these two proteins at the neck of these invaginations.

Certainly, lipid gradients might exist in the vicinity of TMEM16F after activation of scrambling, and we are especially intrigued by this possibility in our new experiments showing the activation of this pathway in control cells during cell swelling. However, it is certain that in the Ca elevation protocols we are generating a systemic loss of membrane asymmetry. Thus, while colocalization of these proteins is attractive, it is not definitively required to explain our data sets available at this time. Structural work to determine how the proteins involved are physically related will be of great interest, but it goes beyond the scope of the present article. For now, therefore, we have strived to improve the discussion of how localized signaling may be important in the present results.

iii. The authors should show that dynamin localization is altered by TMEM16F activation: at rest dynamin localizes at the neck of these invaginations and that upon activation of the scramblase dynamin moves away.

We point out that our hypothesis does not depend on the complete dissociation of dynamin from the necks of the invaginations. Indeed, numerous publications have described dynamin as having many distinct interactions with the membrane and interacting proteins. The constrictions caused by dynamin rings may clearly vary significantly and such changes likely would not be resolvable by light microscopy, possibly not even by EM. We point the reviewer to two illustrative articles describing the diversity in constrictions of multimeric dynamin rings. (Sundborger A. et al., *Cell Reports*, 2014, PMID25088425; Kong, L. et al., *Nature*, 2018, PMID: 30347313). Both articles illustrate the subtle structural differences between various constriction states of dynamin rings. We have improved the discussion section in this regard by stress that complete dissociation of dynamin from the necks of the membrane compartments need not occur. The relaxation need only be large enough for ions to pass through to allow detection by our methods. Physiologically, and over a larger time frame, the compartments are expected to expand outwardly in response to membrane tension and eventually cytoskeletal changes that may occur. Clearly, there are many intriguing possibilities for future studies, but they all exceed the possible scope of the present article.

These experiments are critical to validate and support the proposed mechanism for the involvement of these proteins in PM expansion.

- It is unclear how PS scrambling specifically impairs dynamin binding to the neck of these invaginations. PS is presumably present throughout the inner leaflet of the PM, and TMEM16F activation results in PS externalization throughout the cell. If PS is present throughout the inner leaflet, how does dynamin 'know' where to specifically cluster and organize to form the tight constrictions necessary to isolate these invaginations? Are these compartments particularly enriched in PS? If so, this should be experimentally demonstrated. If not, then what drives dynamin to form these constrictions?

In our experiments and model, dynamin localization to the membrane with compartment formation would occur prior to TMEM16F activation and lipid scrambling. That said, the reviewer is correct in that dynamin localization to specific regions of the cell has been the subject of many investigations over decades and likely relies on additional adapter proteins such as amphiphysin or endophilin. Likewise, it is still not established that lateral inner leaflet phospholipid gradients exist or are a key factor, even for conventional endocytosis. In short, it exceeds a reasonable scope for this article to address in more detail the issue of dynamin localization and lateral organization of inner leaflet anionic lipids. Dynamin definitively forms stable multimeric ring states, binding to the inner membrane with a clear dependence on anionic lipids such as PA, PS, and PI. 'Clustering' of dynamin may or may not involve 'clustering' of anionic lipids. Our present results make a case that loss of anionic lipids is importantly involved in relaxation of dynamins, but this is only one part of the mechanism. Our new results with phosphatidylcholine clearly indicate that additional membrane changes are important to open the compartments, and we have revised our hypotheses accordingly. Elucidation of further details requires much further work that exceeds the present scope.

- Are cells lacking TMEM16F (and/or dynamin) more susceptible to hypotonic stress? The authors' conclusions on the role of these proteins in PM expansion process would predict that the response of cells lacking these proteins to hypotonic stress should be impaired. This should be tested.

The reviewer raises a very good point and after significant experimentation we have now demonstrated that cell swelling through hypotonic stress indeed activates TMEM16F dependent scrambling (Fig. 4). In cells lacking TMEM16F there is still Ca entry through an osmotic or mechanosensitive Ca entry pathway, but PS-specific scrambling does not occur, as indicated by Annexin V staining. These experiments actually demonstrate two principles. A) Hypotonic stress activates TMEM16F dependent scrambling, and B) Ca entry during cell swelling does not conduct through TMEM16F channels. Work of the Cahalan group described these processes, the Ca influx pathway being dubbed SWAC (swell-activated calcium channels; Ross and Cahalan, *JGP*, 1995, PMID: 8786341). As already noted, the SWAC channels are still not identified with confidence. It is certainly possible at this point that multiple Ca-permeable mechanosensitive channels are involved, including PIEZO isoforms. In conclusion, we concur with all of these comments, and the new experiments have added considerably to our study. However, further work to analyze cell survival after swelling exceeds the reasonable scope of this article. We do agree that there should be a marked difference in such survival experiments. We have tried to improve the discussion to reflect all of these points.

- The authors claim to identify the PM compartment that is opened by upon activation of TMEM16F and dynamins as the enlargeosomes. However, this claim is not well founded in the results. A more direct comparison between the properties of enlargeosomes and the compartments identified here is needed to draw the identification drawn by the authors. The evidence that VAMP4-containing compartments are preferentially fused into the PM is not sufficient to draw firm conclusions as this marker is not only present in enlargeosomes. Rather, VAMP4 is associated with the trans-Golgi network and in early endosomes, which argues against the authors' conclusion that PM expansion is not mediated by endosome fusion.

We agree that the literature concerning enlargeosomes is not definitive in multiple respects. However, that literature is clearly relevant and required some level of attention in this study. Thus, we tested whether expressed VAMP4 might colocalize to the compartments that open. Beyond this, we make no further claims about enlargeosomes and concur with the reviewer's statements. We have modified our text accordingly. The claimed relations between VAMP4, the TGN and EE are presumably correct. But our results show that these compartments do not contribute to the newly exposed membrane in an immediate fashion. It is also a question whether overexpression of VAMP4 might misdirect membrane and give a false impression. In any case, the classical membrane compartments cannot constitute the majority of the membrane signal we see, and we have clarified this further in the revised manuscript. Multiple Golgi markers, recycled membranes, ER, and lysosomes do not contribute significantly to the increases in cell surface area (Supp. Fig. 4/Sup. Vid. 6) as seen via TB colocalization, quenching and fluorescence loss experiments.

- The authors' data suggests that Ca²⁺ entry through PIEZO1 can play a role in activating TMEM16F. However, the experiments described here were performed under overexpression conditions casting serious doubts on whether this occurs in physiological conditions, as claimed

by the authors, or if it is due to the artificial experimental set up. The authors should repeat these experiments in cells that endogenously express both PIEZO1 and TMEM16F.

We are in agreement with the reviewer (and Reviewer#2) that the physiological Ca-influx mechanism that promotes TMEM16F activity during cell swelling may include cation channels besides PIEZO1 and we have made modifications to the model accordingly. To be clear, expression of PIEZO1 can promote the pathway to PM expansion, but the physiological channels that underlie TMEM16F-dependent PM expansion during swelling remain to be determined. PIEZO1 is now used as an example of how this might occur, but other channels are likely physiologically more important.

- I am puzzled by the authors' inclusion of PIEZO1 among the channels that are activated by cell swelling. Neither of the reviews associated with this assertion contain data (or references to) that support this notion. Rather, hypotonic stress activates the LLRC/SWELL channels, which are anion selective and Ca²⁺ impermeable. The authors should rule out the possibility that the effects seen here are mediated by these channels, i.e. showing that the responses seen here are not affected by LLRC-specific inhibitors.

Please see the above comments regarding PIEZO1. That said, expressed PIEZO channels CAN clearly be activated during membrane tension changes that occur during hypotonic stress. The group of Ardem Patapoutian first identified PIEZO as a mechanosensitive channel using assays that induce osmotic stress with our current contributor R. Syeda, performing the majority of the assays with purified proteins and lipid bilayers (R. Syeda, et. al., *Cell Reports*, 2016, PMID: 27829145). In the present study, we have pharmacologically activated PIEZO1 using the specific agonist Yoda1 to rule out any effect of LLRC/SWELL channels. We have also eliminated TMEM16F as the source of Ca entry during osmotic stress. The elimination of chloride from our buffers does not block PM expansion (Bricogne, C. et. al., *Scientific Reports*, 2019, PMID: 31097725) and therefore it is unlikely that LLRC/SWELL contributes to the Ca-entry either directly or upstream in our responses. We now clarify this point in the discussion.

- Is the capacitance of cells lacking dynamin and/or TMEM16F constitutively higher than that of WT cells? If not, what happens to the excess membrane? Is it not synthesized?

Yes, as mentioned above, we now include this data in Figure 1. The excess PM area of Dnm TKO MEF cells (i.e., the ratio of real empirical C_m based membrane area to spherical cell area) is definitively larger than their non-induced counterparts. We suggest that this may be in part because the compartments analyzed are already open or, of course, the disruption of conventional endocytic processes involving dynamin may also play a role in these phenomena although several new experiments now included involving K44A mutations of Dnm2 suggest this may not be the case. As described for reviewer #1, the previous EM work performed by the De Camilli group reveals an expansive network of exposed compartments described as "tubulated pits" and "tubulated fragments", possibly representing the membrane described in our paper. However, both hypotheses are now addressed and the inclusion of the new data supporting the cell area/capacitance is included in the supplement and the inclusion of this into this version of the paper has added significantly to the quality and understanding of how dynamin is involved in the observed PM expansion.

REVIEWER COMMENTS

Reviewer #1 (Remarks to the Author):

Deisl et al have addressed my questions. The work is thorough and conclusions justified by the experiments. Many questions remain about control of plasma membrane area, and this paper puts forth a well justified model.

Reviewer #2 (Remarks to the Author):

The authors have addressed many of my previous concerns and have made significant improvements. I am finally beginning to believe their story. Nevertheless, there are a few important holes that need to be filled.

Major:

My main scientific concern regards data supporting the model in Figure 5. There are two key pieces of data that are missing to bring this model home. (1) The authors propose that binding of dynamin to acidic phospholipids is responsible for the constriction. Mutation of the $\beta 6$ - $\beta 7$ loop in the PH domain eliminates phospholipid binding. The ability of this mutant to rescue the TKO cells should be shown. Although the authors might balk at this suggestion with the argument that the effect of PC tends to minimize (or at least complicate) the role of dynamin binding to anionic lipids, the authors suggest that the localization of dynamin to the neck is effected by anionic lipids. Therefore, this experiment is critical. (2) The authors propose that dynamin is responsible for the constriction of the membrane compartment, however, they present no evidence that dynamin is localized there. They should show super-resolution images of immune-stained or overexpressed fluorescent dynamin demonstrating that dynamin is actually located at the constriction. In my opinion, these two pieces of data are essential for this hypothesis. These two experiments would convince me.

Maybe it is just me, but I feel the presentation still leaves a lot to be desired. There is something about the writing style and the multi-multi part figures that made it necessary for me to spend inordinate amounts of time working through each figure. For example, Figure 1C alone involves 6 different variables: (1) induced and uninduced, (2) stimulated with Ca or not stimulated with Ca, (3) temperature, (4) transfection with K44A, (5) HEK and Dnm TKO cells, and (6) Cm and excess PM. OK, it finally all makes sense when I sort it out, but the way it is packaged makes my head spin. In Figure 3, the authors apparently have tried to help the reader by including little cartoons, but the cartoons are not very helpful. In A, for example, it looks like PS and PA are being perfused. The use of 3 different cell types in this figure (and more than that in the paper) does not help. If the authors could restrict their data in the main body of the paper to one – or maybe 2 – cell lines and reserve all the other cell types to supplementary data, this would eliminate my trying to keep straight the differences between the different cell types and what experiments have been done in what cell type.

Random minor annoyances:

P4. “multiple mutations in TMEM16F that alter ion selectivity and activation kinetics but maintain sufficient PL scrambling”. Maybe I am missing something, but ref 17 shows that the Q559K mutant

has diminished PS exposure (~half of WT). And, the only other mutant in this paper is D430G, which the authors report has normal PS exposure.

P7. "using highly Ca buffered cytoplasmic solutions (free Ca, 7 μ M)". EGTA cannot buffer Ca at this concentration.

Fig. 1C, Dnm TKO label is misplaced.

Why do the authors show the effect of the K44A mutant at 23C rather than at 37C?

These sentences are enigmatic to me: "Importantly, the decrease of excess PM in HEK cells is of large magnitude and is opposite to the expectation that K44A mutant phenotypes can be explained by inhibition of endocytosis. In fact, the generation of tubular membrane structures was noted previously."

P8. The authors state: "The fact that Dnm deletions do not affect PL scrambling or MEND responses is consistent with Dnm directly regulating PM expansion, immediately downstream of PL scrambling." I think that the data preceding this sentence do not imply any connection between PM expansion and PL scrambling.

As I noted previously, the Amph-SH3 experiments seem incomplete - they lack controls and the mechanisms remain obscure.

Reviewer #3 (Remarks to the Author):

I find the manuscript by Deisl and colleagues much improved. The additional data provided is very interesting and overall strengthens the conclusions. The extensive rewriting has clarified many of the confusing points I raised earlier.

One critical remaining point is that the authors show (Fig. 4) that hypotonic-induced cell-swelling activates TMEM16F-mediated scrambling, likely because SWAC channels activation allow Ca²⁺ entry which in turn promotes scramblase activity. The authors report that in hypotonic conditions only ~20% of the cells externalize PS (Fig. 4B bottom panel). Does this correlate with the fraction of cells that swells? If so, this should be reported. If not, then how do the authors explain the incomplete response? In the remaining 80% of the cells the PM expands under the hypotonic stress, but scrambling does not occur. This would argue that TMEM16F is not necessary for the process. Ideally the authors should address this issue experimentally. One possibility is that there is some cell-to-cell variability in the kinetics of PS externalization and that the authors did not wait long enough for sufficient amounts of PS to appear on the outer leaflet to ensure a detectable Annexin V signal. This could be readily testable by waiting for longer times and seeing if the fraction of scrambling positive cells increases. If this is not possible, then the discrepancy should be acknowledged and a possible explanation provided.

A minor note is that I find the figure organization very difficult to follow. The authors group multiple

figure panels under a single letter, making the life of a reader more difficult than necessary. For example Fig. 3D has a scheme, 2 DeltaCm traces, 3 images of cells and 2 summary bar graphs. I would suggest that if the various groups were indicated with different letters it would be easier to follow the description of the data in the text.

RESPONSE TO THE REVIEWS.

TMEM16F and dynamins control expansive plasma membrane reservoirs.

We provide the following point-point response to the reviewers' comments:

Reviewer #1 (Remarks to the Author):

Deisl et al have addressed my questions. The work is thorough and conclusions justified by the experiments. Many questions remain about control of plasma membrane area, and this paper puts forth a well justified model.

We appreciate this kind response and the helpful reviews that reviewer #1 has provided for the development of our publication. Indeed the latest version addresses substantially more definitively the outstanding questions and provides a clearer path for future work within this field.

Reviewer #2 (Remarks to the Author):

The authors have addressed many of my previous concerns and have made significant improvements. I am finally beginning to believe their story. Nevertheless, there are a few important holes that need to be filled.

We are thankful that the previous improvements to our manuscript have been noticed and appreciate the thoughtful feedback provided by Review#2. In fact, our gratitude could not be greater, since the reviewers comments motivated us to carry out entirely new types of experiments that have now qualitatively elevated the manuscript. We believe the fundamental progress achieved in the revision will be entirely evident without expanding extensively on the outcomes here. In short, dynamins definitively are released from the membrane in parallel with PL scrambling and with very strong certainty mediate the opening of membrane compartments that otherwise would have been interpreted as exocytic compartments. We are confident this progress will have major cell biological implications across all fields that touch on membrane biology.

Major:

My main scientific concern regards data supporting the model in Figure 5. There are two key pieces of data that are missing to bring this model home.

- (1) The authors propose that binding of dynamin to acidic phospholipids is responsible for the constriction. Mutation of the $\beta 6$ - $\beta 7$ loop in the PH domain eliminates phospholipid binding. The ability of this mutant to rescue the TKO cells should be shown. Although the authors might balk at this suggestion with the argument that the effect of PC tends to minimize (or at least complicate) the role of dynamin binding to anionic lipids, the authors suggest that the localization of dynamin to the neck is effected by anionic lipids. Therefore, this experiment is critical.

We believe reviewer #2 is referencing the K562E Marie Charcot Tooth syndrome mutation in dynamin2. This disease causing mutation has been found to abolish anionic lipid binding in the PH domain of dynamin2. We had initially planned to include this clinically relevant mutation in subsequent work but in response to this review we analyzed the role of K562E in scrambling dependent PM expansion. As the reviewer noted, it was difficult to hypothesize whether anionic lipid binding mutants would have a significant impact on dynamin localization after scrambling. To our surprise, the K562E had a resoundingly different phenotype than the K44A GTPase dead and WT dynamin2 expression. We now include a significant new data set visualizing and quantifying how both K44A and WT can rescue dynamin dependent compartment formation and scrambling dependent expansion while the anionic lipid binding mutant fails to rescue. These experiments also allowed super-resolution visualization of dynamin punctae at the membrane adhesion interface where again to our surprise, Ca activated a dramatic dissipation in dynamin punctae for WT and K44A in a TMEM16F-dependent manner but failed to do so for the K562E mutation. Taken together these data clearly point to remarkable control of the membrane reserve by dynamin through its binding of anionic lipids which is disrupted by TMEM16F dependent scrambling. The reviewer is correct that the role of PC and particular roles of specific anionic lipids is complicated and future work detailing these responses will need to be performed. We maintained only part of this data set in Figure 3, accordingly, simplifying it and providing extensive analysis in the supplement to allow for new highlighting of dynamin punctae and their striking PL scrambling-dependent dissipation from the PM.

- (2) The authors propose that dynamin is responsible for the constriction of the membrane compartment, however, they present no evidence that dynamin is localized there. They should show super-resolution images of immune-stained or overexpressed fluorescent dynamin demonstrating that dynamin is actually located at the constriction. In my opinion, these two pieces of data are essential for this hypothesis. These two experiments would convince me.

Please see above.

Maybe it is just me, but I feel the presentation still leaves a lot to be desired. There is something about the writing style and the multi-multi part figures that made it necessary for me to spend inordinate amounts of time working through each figure. For example, Figure 1C alone involves 6 different variables: (1) induced and uninduced, (2) stimulated with Ca or not stimulated with Ca, (3) temperature, (4) transfection with K44A, (5) HEK and Dnm TKO cells, and (6) Cm and excess PM. OK, it finally all makes sense when I sort it out, but the way it is packaged makes my head spin. In Figure 3, the authors apparently have tried to help the reader by including little cartoons, but the cartoons are not very helpful. In A, for example, it looks like PS and PA are being perfused. The use of 3 different cell types in this figure (and more than that in the paper) does not help. If the authors could restrict their data in the main body of the paper to one – or maybe 2 – cell lines and reserve all the other cell types to supplementary data, this would eliminate my trying to keep straight the differences between the different cell types and what experiments have been done in what cell type.

We understand that there is a significant amount of data we are trying to present in such limited space and have made significant progress in streamlining this in our current revision. Much of this is helped by the reviewers previous experimental suggestions which have significantly helped us improve presentation of the take home message. In particular, for Figure 1, we have eliminated many redundant descriptions across multiple lines or with multiple controls and placed them into the supplement. I have reworked the design of the figures to improve visualization and reduce redundancies to the supplement as suggested. We have removed the cartoons and simplified Figure 3, minimizing the cell types to just 2. The main cell types utilized for this paper are primarily 1) Pietro De Camilli's MEF Dynamin TKO cell line due to the ability to remove all three isoforms of dynamin. 2) To a lesser extent, the HEK cell line due to the near ubiquitous nature of this system and amenability to patch clamping (In addition, the HEK cell has been well characterized as having little to no background PIEZO activity). 3) Jurkat cells as they have been previously characterized in TMEM16F mediated membrane expansion and their history to induce osmotically induced Ca-currents was established over 30 years ago with the discovery of SWAC channels. 4) BHK-NCX cells, BHK cells provide an alternative method for ionophore independent rapid cytoplasmic Ca manipulation as well as providing robust visualization during combined electrophysiology and confocal experiments. We have chosen to reduce significantly the complication of additional cell lines and focus primarily on the MEF cells with additional physiological examinations being slightly more cell type specific. Our initial intent was to show the diversity of these responses across multiple cell lines but do indeed agree with the reviewer. Now that the overall paper is more focused, and more importantly contains ground-breaking new data about dynamin-membrane interactions (promoted by Reviewer #2), these duplicate data sets can be assigned to the supplementation.

Random minor annoyances:

P4. "multiple mutations in TMEM16F that alter ion selectivity and activation kinetics but maintain sufficient PL scrambling". Maybe I am missing something, but ref 17 shows that the Q559K mutant has diminished PS exposure (~half of WT). And, the only other mutant in this paper is D430G, which the authors report has normal PS exposure.

This was a slight grammatical confusion. There were multiple mutations in TMEM16F that altered either ion selectivity of activation kinetics. We have corrected the sentence to avoid confusion by removing 'Multiple' and inserting 'either'; "Mutations in TMEM16F that either alter ion selectivity or activation kinetics maintain sufficient PL scrambling...."

P7. "using highly Ca buffered cytoplasmic solutions (free Ca, 7 μ M)". EGTA cannot buffer Ca at this concentration.

This is a common problem, certainly, and the comment is correct in principle. EGTA is a poor buffer at 7 μ M free Ca. At pH 7.0, the solution contains 9 mM Ca and 10 mM EGTA. Still, the buffer capacity is >10 to 1. A 100 μ M change of total Ca results in a 6 to 7 μ M change of free Ca. Being well aware of buffer biophysics, we also performed many experiments using other buffers with low affinity, e.g. citrate and NTA. Results were not different, but we choose not to present for reasons of brevity.

Fig. 1C, Dnm TKO label is misplaced.

Corrected.

Why do the authors show the effect of the K44A mutant at 23C rather than at 37C?

For analysis of PM expansion and all optical analysis K44A followed the same 37C as all other variables. For Fig 1C where excess basal membrane area was analyzed without Ca we performed all the experiments at RT to minimize any background membrane dynamics and Ca flux. We have clarified this in the text explaining the difference and why it was performed this way.

These sentences are enigmatic to me: "Importantly, the decrease of excess PM in HEK cells is of large magnitude and is opposite to the expectation that K44A mutant phenotypes can be explained by inhibition of endocytosis. In fact, the generation of tubular membrane structures was noted previously."

Corrected.

P8. The authors state: "The fact that Dnm deletions do not affect PL scrambling or MEND responses is consistent with Dnm directly regulating PM expansion, immediately downstream of PL scrambling." I think that the data preceding this sentence do not imply any connection between PM expansion and PL scrambling.

Corrected

As I noted previously, the Amph-SH3 experiments seem incomplete - they lack controls and the mechanisms remain obscure.

Due to the inclusion of additional acute dynamin visual experiments at the suggestion of the reviewer (for which we are most grateful), we have placed the Amph-SH3 experiments into the supplemental data. That said, the data supports the notion transient Ca-stimulation of PM expansion is blocked when dynamin is bound to its high affinity amphiphysin SH3 domain partner. This could prevent dynamin relaxation at the membrane interface. Regarding the controls, both data sets and multiple cell types were tested with their purified protein buffer controls. Additionally, dialysis of smaller peptides and antibodies has long been characterized for electrophysiological analysis with pipette resistances in the 2-10 MOhm range. Figure 3 has time-resolved optical verification of fluorescently tagged secondary labeled antibodies dialyzing into the cytoplasm as an additional example.

Reviewer #3 (Remarks to the Author):

I find the manuscript by Deisl and colleagues much improved. The additional data provided is very interesting and overall strengthens the conclusions. The extensive rewriting has clarified many of the confusing points I raised earlier.

Thank you for your patience understanding and valued input.

One critical remaining point is that the authors show (Fig. 4) that hypotonic-induced cell-swelling activates TMEM16F-mediated scrambling, likely because SWAC channels activation allow Ca^{2+} entry which in turn promotes scramblase activity. The authors report that in hypotonic conditions only ~20% of the cells externalize PS (Fig. 4B bottom panel). Does this correlate with the fraction of cells that swells? If so, this should be reported. If not, then how do the authors explain the incomplete response? In the remaining 80% of the cells the PM expands under the hypotonic stress, but scrambling does not occur. This would argue that TMEM16F is not necessary for the process. Ideally the authors should address this issue experimentally. One possibility is that there is some cell-to-cell variability in the kinetics of PS externalization and that the authors did not wait long enough for sufficient amounts of PS to appear on the outer leaflet to ensure a detectable Annexin V signal. This could be readily testable by waiting for longer times and seeing if the fraction of scrambling positive cells increases. If this is not possible, then the discrepancy should be acknowledged and a possible explanation provided.

The reviewer raises some interesting points. To clarify, the number of cells responding to Annexin5 is the number of cells labeled with a discrete membrane labeling of AnnexinV per image over total number of cells per image. Measurements of Ca using xRhod-1 were transient in cells with not all cells responding at once. To address this, we monitored Ca level independent of Annexin staining and counted all cells that responded to osmotic stress measuring their average maximum and basal response during the experiment over time. Indeed, when Ca was monitored during hypotonic stress, more cells responded in both groups, as one would expect if the Ca entry source was NOT TMEM16F and a yet to be identified osmosensitive SWAC channel. Both groups responded to osmotic tension relatively similarly indicating that TMEM16F expression did not change how much Ca enters the cytoplasm during hypotonic stress. As for why only a maximum of 25% of the cells were stained with Annexin V, I believe the reviewer is correct. Over a longer period of time more cells would respond. Annexin V does not respond to PS exposure as quickly as K7r. However, if cells were treated with hypotonic solution for an extended period of time cell lysis begins to occur with AnnexinV staining lysed cells as well as PS exposure. While this experiment is an interesting a measurement of cell lysis in hypotonic conditions as controlled by TMEM16F expression this goes beyond what is necessary to illustrate in this proof of concept figure. Subsequent follow-up work focusing more on the physiology of membrane stress and TMEM16F scrambling induced PM expansion should detail these responses and time scales.

A minor note is that I find the figure organization very difficult to follow. The authors group multiple figure panels under a single letter, making the life of a reader more difficult than necessary. For example Fig. 3D has a scheme, 2 DeltaCm traces, 3 images of cells and 2 summary bar graphs. I would suggest that if the various groups were indicated with different letters it would be easier to follow the description of the data in the text.

Corrected as suggested. See above, Figure 3 has been significantly focused and simplified.

REVIEWERS' COMMENTS

Reviewer #3 (Remarks to the Author):

The authors have addressed all my concerns. I have no further requests of them. This is a very interesting manuscript that will open new and exciting lines of investigation into the role of the TMEM16 scramblases and their involvement in regulating cellular processes.